# AAV11 enables efficient retrograde targeting of projection neurons and enhances astrocyte-directed transduction

Zengpeng Han [1,2,3,4,10], Nengsong Luo [5,10], Wenyu Ma [2,4], Xiaodong Liu[6], Yuxiang Cai[5], Jiaxin Kou [7], Jie Wang[2,4], Lei Li[2], Siqi Peng [8], Zihong Xu [8], Wen Zhang [8], Yuxiang Qiu[1,3,4], Yang Wu[2,4], Chaohui Ye[2,4], Kunzhang Lin [1,3] ✉ & Fuqiang Xu [1,2,3,4,5,9] ✉

Viral tracers that enable efficient retrograde labeling of projection neurons are powerful vehicles for structural and functional dissections of the neural circuit and for the treatment of brain diseases. Currently, some recombinant adeno-associated viruses (rAAVs) based on capsid engineering are widely used for retrograde tracing, but display undesirable brain area selectivity due to inefficient retrograde transduction in certain neural connections. Here we developed an easily editable toolkit to produce high titer AAV11 and demonstrated that it exhibits potent and stringent retrograde labeling of projection neurons in adult male wild-type or Cre transgenic mice. AAV11 can function as a powerful retrograde viral tracer complementary to AAV2-retro in multiple neural connections. In combination with fiber photometry, AAV11 can be used to monitor neuronal activities in the functional network by retrograde delivering calcium-sensitive indicator under the control of a neuron-specific promoter or the Cre-*lox* system. Furthermore, we showed that GfaABC$_1$D promoter embedding AAV11 is superior to AAV8 and AAV5 in astrocytic tropism in vivo, combined with bidirectional multi-vector axoastrocytic labeling, AAV11 can be used to study neuron-astrocyte connection. Finally, we showed that AAV11 allows for analyzing circuit connectivity difference in the brains of the Alzheimer's disease and control mice. These properties make AAV11 a promising tool for mapping and manipulating neural circuits and for gene therapy of some neurological and neurodegenerative disorders.

Structural changes or loss of neural circuits are attributed to several major neurological diseases, such as Alzheimer's disease, Parkinson's disease, schizophrenia, autism, etc., imposing a heavy burden to the family and society[1]. Analyzing the structure and function of brain neural network is fundamental to revealing the working principle of brain and the mechanism of brain diseases[2,3]. However, mapping and manipulating functional network connection has become an important challenge due to the lack of available techniques for targeting projection neurons. While classic neuroanatomical retrograde tracers are widely used to analyze the architectural connectivity between different brain regions, they cannot deliver genetic cargos for neural activity manipulation and clinical application[4]. Recombinant viral vectors have been developed as valuable tools to address the limitations of traditional retrograde tracers, allowing sufficient payload expression for circuit dissection and in vivo gene therapy[5].

Many naturally evolved and engineered viral vectors can transduce projection neuron populations by axon terminal uptake, including adeno-associated virus (AAV)[6–8], canine adenovirus-2 (CAV-2)[9], rabies virus (RV)[10], herpes simplex virus (HSV)[11], and lentivirus (LV)[12], among others. Of these, CAV-2 can effectively retrograde label long-range projection neurons[9], but its modification and preparation are not as straightforward as AAV[13]. While RV and HSV display robust efficiencies of retrograde labeling with rapid gene expression, they are not suitable for functional research and gene therapy due to high cytotoxicity[10,11]. Lentiviruses pseudotyped with modified RVG present the improved ability for retrograde transduction in rodents and non-human primates[12,14], but have a risk of tumorigenesis due to random integration into the host genome[15]. Using engineered AAVs to target projection neurons is a priority because they have some unique advantages, including low pathogenicity, long-term gene expression, extensive tissue tropisms and approved clinical treatment cases[5,16].

Several engineered AAVs, such as AAV2-retro[7], AAV9-retro[6], and AAV2-MNM004[8], which were developed through directed evolution or rational design and exhibit robust retrograde transport, have been extensively used to express fluorescent probes to analyze the structural connections of neural networks and to express functional molecules (e.g., indicators or effectors of neural activity) to monitor and manipulate neuronal activities[7]. However, these viral vectors display unwanted brain area selectivity due to inefficient retrograde transduction in certain neural connections[7,10]. In addition to the capsid engineering methods (e.g., directed evolution or rational design), an alternative approach to find a better AAV is to identify new natural serotypes or uncover new properties of natural serotypes[17]. There are many natural serotypes of AAV, which belong to different clades and have their unique infectious characteristics[17–19]. In neuroscience research, the commonly used AAV serotypes are 1, 2, 5, 6, 8, and 9[20], but low propensity of retrograde transduction hinders their applications in the research of projection networks or the treatment of disease. Therefore, it is crucial to persist in screening versatile AAVs capable of retrograde labeling projection neurons with high efficiency.

Here, we screened AAVs that have not been widely used in neuroscience research and found that AAV11 exhibits potent retrograde labeling of projection neurons with enhanced efficiency over AAV2-retro in some neural connections. The AAV11 can be utilized alone or in combination with the Cre recombinase-lox system to achieve high-level gene expression for functional monitoring of neuronal activities. In addition, we showed the competence of AAV11 for astrocyte targeting with an efficiency significantly higher than that of the commonly used AAV8 and AAV5. Combined with bidirectional multi-vector axoastrocytic labeling, AAV11 can be applied to study neuron-astrocyte connection. Finally, we showed that AAV11 allows for dissecting circuit connectivity difference in the brains of the Alzheimer's disease and control mice. These properties make AAV11 a valuable option for the mapping and manipulating of neural circuits and for gene therapy of some neurological and neurodegenerative disorders.

## Results

### AAV11 mediates efficient neuron transduction with axon terminal absorption

For the practical production of AAV11 vectors, we established a triple plasmid system for AAV11 packaging. Subsequently, the efficiency of virus packaging was evaluated in HEK-293T cells. We found that the pAAV2/11 plasmid could be used to package high-titer AAV11 and the yields of viral particles was equivalent to those of AAV9 (Supplementary Fig. 1).

To evaluate the transduction effect of AAV11, AAV11-EF1α-EGFP virus was infused into the caudate-putamen (CPu) of adult mice via stereotactic injection, and then CPu and the well-characterized upstream regions projecting to the CPu were imaged at 21 days post-injection (DPI) (Fig. 1a). We found a broad spread of EGFP expression in situ and upstream of CPu (Fig. 1b-f), including the primary somatosensory area (SSp) and the mediodorsal nucleus of thalamus (MD), among others. Consistently, we injected AAV11-EGFP and AAV2-retro-mCherry in equal amounts into the CPu and found that AAV11 and AAV2-retro could retrogradely co-label the upstream brain regions of the CPu (Supplementary Fig. 2). Even though the in situ diffusion range of AAV11 was reduced by reducing the injection dosage, AAV11 still showed strong retrograde targeting of projection neurons in MD (Supplementary Fig. 2c and d). In addition, similar brain-wide retrograde labeling by AAV2-retro and AAV11 were also observed using another strong promoter CAG (Supplementary Fig. 3a). We evaluated the transduction efficiency of the AAV11 virus in the substantia nigra pars compacta (SNc) to dorsal lateral striatum (DLS) pathway, and found that its retrograde transduction efficiency was comparable to that of AAV2-retro (Supplementary Fig. 3b).

Some AAV serotypes (such as AAV1 and AAV9) exhibit the features of anterograde transsynaptic propagation[21], which can also cause an extensive spread. To test whether AAV11 can mediate anterograde transsynaptic spread, AAV11-hSyn-Cre was injected into primary visual cortex (V1) of Ai14 (CAG promoter-driven and Cre-dependent expression of tdTomato reporter) mice[22] (Supplementary Fig. 4a). After 3 weeks post-injection, extensive tdTomato fluorescence signals were detected at the injection site, but no tdTomato-expressing cell bodies were observed in the downstream regions known to be directly projected by V1 (Supplementary Fig. 4b–e), including the superior colliculus (SC) and the CPu[23,24], indicating that AAV11 does not spread anterograde across synapses in primary visual cortex. These results suggested that AAV11 can be retrogradely transported to the cell bodies of projection neurons through axon terminal uptake.

### AAV11 can trace projection neurons that AAV2-retro does not easily transduce

As an effective and practical retrograde viral tracer, AAV2-retro is widely used in the analysis and manipulation of different types of neural circuits[25] as well as disease modeling[26]. In the experiments mentioned above, when initiating retrograde labeling of upstream brain areas from CPu, AAV11 was able to label most of the same brain areas as AAV2-retro. However, in some upstream brain areas, AAV11 and AAV2-retro exhibited different neuronal subpopulation targeting. For example, in the cortical SSp area, they tended to label different layers (Supplementary Fig. 2b). To further assess the performance of AAV11 retrograde tagging, AAV11-EF1α-EGFP and AAV2-retro-EF1α-mCherry were mixed in equal amounts and injected into the ventral hippocampus (vHPC) (Fig. 2) or the periaqueductal gray (PAG) (Fig. 3). When injected into the vHPC area (Fig. 2a), both viruses can label a large number of neurons at the injection site (Fig. 2b). AAV11 can strongly label the projection neurons from the hippocampal field CA3 (CA3), the nucleus of reuniens (RE), the endopiriform nucleus dorsal part (EPd), the lateral dorsal nucleus of thalamus (LD), and the medial septal complex (MSC), while to a significantly lower extent, AAV2-retro labeled only a fraction of projection neurons in the same regions (Fig. 2c, d). Quantitative analysis showed that AAV2-retro was more prone to retrograde infect the entorhinal cortex (ENT), with a significantly higher transduction efficiency than that of AAV11 ($p = 0.0205$, Fig. 2e). In contrast, AAV11 has higher retrograde transport capacity than AAV2-retro in other upstream brain regions including the dorsal hippocampus (dHPC, $p = 0.0033$), the midline group of the dorsal thalamus (MTN, $p = 0.0048$), the EPd ($p = 0.0015$), the LD ($p = 0.0027$), and the MSC ($p = 0.0357$) (Fig. 2e). Importantly, similar effects were observed in the experimental groups of separate injection (Supplementary Fig. 5), different virus batches (Supplementary Fig. 6) and color exchange (Supplementary Fig. 7). Even though the injection dosage of AAV11 at vHPC was greatly reduced, it could still retrograde target the projection neurons in EPd, while AAV2-retro mainly targeted the projection neurons in piriform area (PIR) (Supplementary Fig. 8).

The PAG is a critical structure for pain modulation, sympathetic responses, and the learning and acting of defensive and aversive behaviors[27]. We tested the ability of AAV11 and AAV2-retro to retrograde label the upstream brain regions of PAG (Fig. 3a, b). We found that there was a comparable tracing effect in the ectorhinal area (ECT) between two viruses (Fig. 3c), while considerably more AAV11-transduced cell bodies were observed in ventromedial hypothalamic nucleus (VMH) (Fig. 3d). Quantitative analysis demonstrated that AAV11 was more efficient than AAV2-retro at retrograde labeling of the central amygdalar nucleus (CEA, $p < 0.0001$), the cerebral cortex layer 6 (CTX L6, $p < 0.0001$), the dorsal premammillary nucleus (PMd, $p < 0.0001$), the substantia innominata (SI, $p < 0.0001$), and the VMH ($p < 0.0001$) (Fig. 3e). In addition, there were similar transduction efficiencies between two serotypes in the cerebral cortex layer 4 and layer 5 (CTX L4, $p = 0.5668$; CTX L5, $p = 0.5914$) (Fig. 3e).

Taken together, these results indicated that AAV11 can function as a powerful retrograde viral tracer, complementary or superior to AAV2-retro.

### Retrograde targeting of genetically defined neuronal populations

We also evaluated whether AAV11 works efficiently with the Cre-lox system by using the Cre transgenic lines for retrograde targeting of genetically defined projection neurons. Specifically, nucleus accumbens (NAc) projecting GABAergic neurons were assessed. GABAergic neurons play versatile roles in various behavioral functions such as reward[28], reinforcement[28], feeding[29] and binge-like alcohol drinking[30] etc. We used an AAV11 vector carrying lox sites flanked nuclear-localized tdTomato, which express the reporter in a Cre-dependent manner (Supplementary Fig. 9a). The vector was injected into the NAc of a GABAergic neuron-specific Cre driver line mice, Vgat-ires-Cre[31] (Supplementary Fig. 9b). After 21 days, multiple brain regions known

to consist of NAc-projecting GABAergic neurons, including lateral hypothalamic area (LHA), striatum-like amygdalar nuclei (sAMY), laterodorsal tegmental nucleus (LDT), and ventral tegmental area (VTA), were positive for nuclear-localized red fluorescent signals (Supplementary Fig. 9c). Consistent with the previous reports[28,29,32,33], abundant cells were labeled in LHA and sAMY, while only a fraction of fluorescent cells were observed in VTA and LDT (Supplementary Fig. 9c). We further compared the efficiency between AAV11 and AAV2-retro in retrograde transduction of GABAergic neurons projecting to NAc (Fig. 4a). We found that there are different tracing effects in the upstream brain regions between two viruses (Fig. 4b). Quantitative analysis demonstrated that AAV11 was more efficient than AAV2-retro at retrograde labeling of the sAMY ($p = 0.0030$) and the PAG ($p = 0.0295$) (Fig. 4c), while AAV2-retro was more efficient than AAV11 at retrograde labeling of the VTA ($p = 0.0256$). In addition, there were similar transduction efficiencies between two serotypes in the dorsomedial nucleus of the hypothalamus (DMH, $p = 0.8956$) (Fig. 4c). These results demonstrated that dissection of genetically defined projection neurons can be implemented by combining AAV11 with available transgenic lines (Cre or Flp).

### Use of AAV11 for functional circuit interrogation

GCaMP can reliably monitor neuronal calcium transients and is widely used to detect the activity of GCaMP-expressing neurons[34]. Dopaminergic projection from the VTA to NAc is activated for work motivation and reward-driven learning[35]. To verify the application of AAV11 in functional circuit interrogation, we tested the dual-viral vector and single-viral vector strategies. For the dual-viral vector strategy, we injected the AAV11-Cre vector into the NAc and simultaneously the Cre-inducible AAV9-GCaMP6m vector into the VTA of C57BL/6 mice (Fig. 5a). Sucrose solution was placed in the behavior box and used as a reward for mice (Fig. 5b). As expected, while mice licked sugar water,

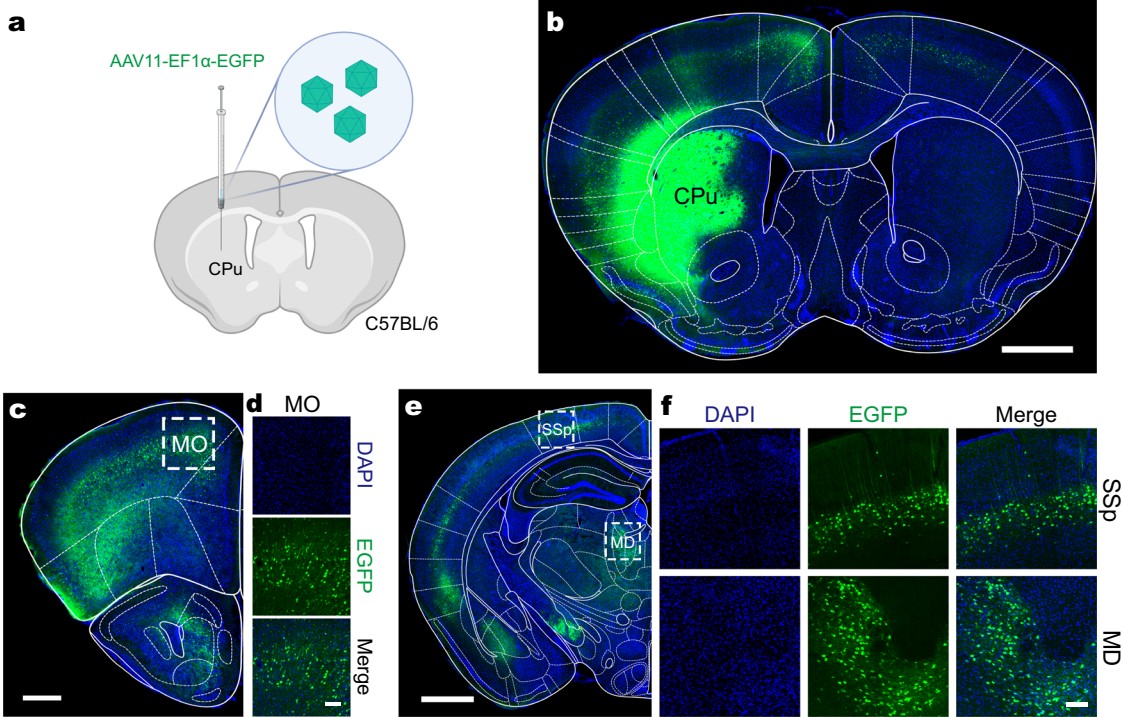

**Fig. 1 | AAV11 for efficient retrograde access to projection neurons. a** Schematic diagram of virus injection. AAV11 ($3 \times 10^9$ vector genomes (VG)) was injected into caudate-putamen (CPu) area of C57BL/6 mice. $n = 3$. Diagram was created with BioRender.com. **b** The fluorescence distribution of EGFP at the injection site of CPu. Scale bar = 1 mm. **c** The spread of the virus to the anterior side. Scale bar = 500 μm. **d** Neurons were labeled in somatomotor areas (MO). Scale bar = 100 μm. **e** The spread of the virus to the posterior side, Scale bar = 1 mm. **f** Neurons were labeled in primary somatosensory area (SSp) and mediodorsal nucleus of thalamus (MD). Scale bar = 100 μm.

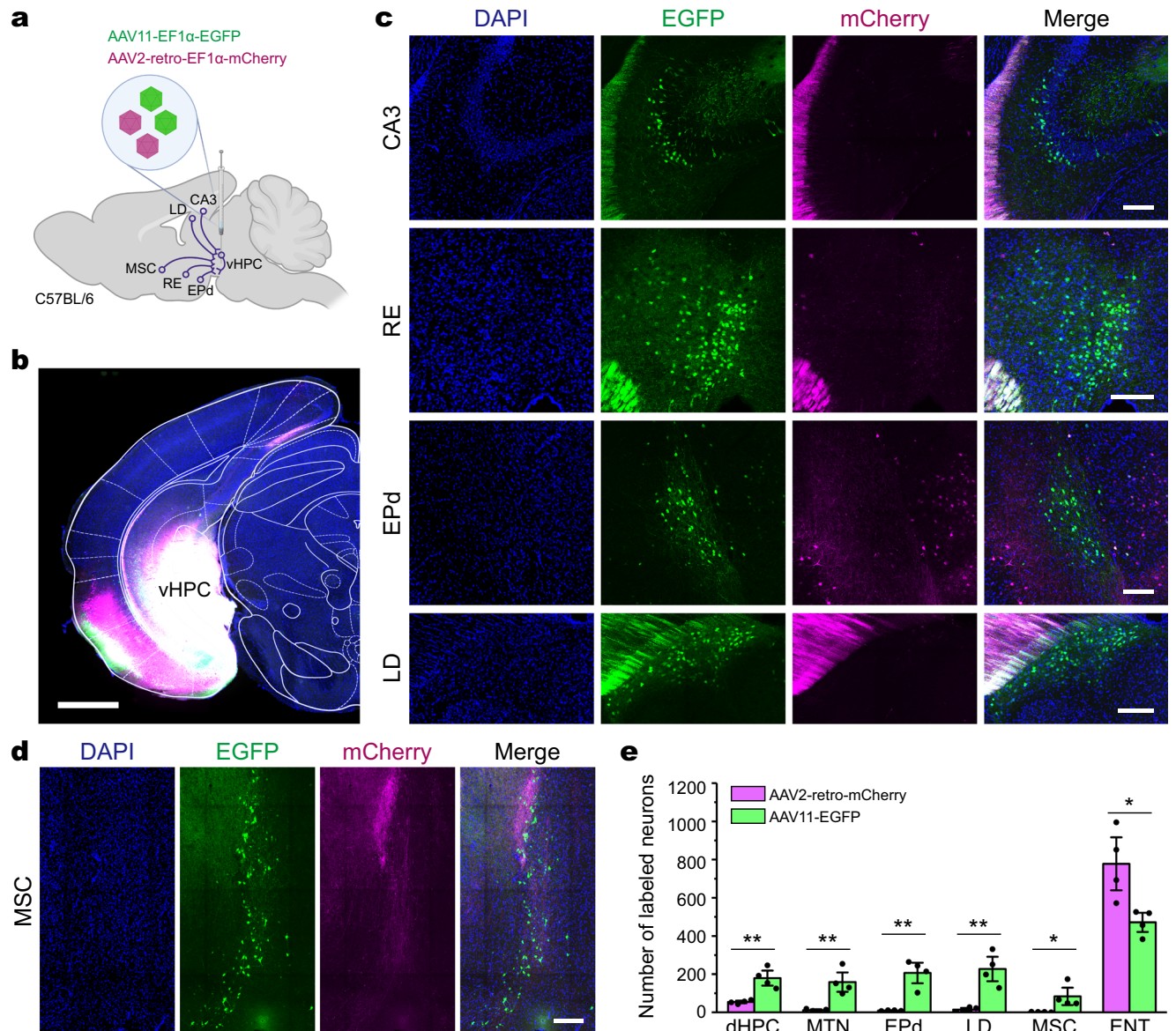

**Fig. 2 | Comparison of AAV11 and AAV2-retro in retrograde transduction tropism after vHPC infusion. a** Connectivity diagram shows the injection site (ventral hippocampus, vHPC) and its upstream areas in hippocampal field CA3 (CA3), nucleus of reuniens (RE), endopiriform nucleus dorsal part (EPd), lateral dorsal nucleus of thalamus (LD), and medial septal complex (MSC). AAV11-EF1α-EGFP and AAV2-retro-EF1α-mCherry viruses were mixed at a particle ratio of 1:1 (3 × 10⁹ VG in total) and injected into vHPC of C57BL/6 mice. Diagram was created with BioRender.com. **b** Fluorescence distribution of EGFP (AAV11) and mCherry

(AAV2-retro) at the vHPC injection site. Scale bar = 1 mm. Representative images reveal that the retrograde labeling efficiencies of AAV11-EF1α-EGFP and AAV2-retro-EF1α-mCherry are quite different in many regions, such as the CA3, RE, EPd, LD (**c**) and MSC (**d**). Scale bar = 150 μm for **c** and 200 μm for **d**. **e** Quantification of AAV11- and AAV2-retro-infected projection neurons. Statistical values are presented as mean ± SEM (*n* = 4/group). Statistical analyses were performed through unpaired two-tailed Student's *t* tests, with significant differences being expressed by the *p* value. \**p* < 0.05, \*\**p* < 0.01. Source data are provided as a Source Data file.

NAc-projecting VTA neurons were activated (Fig. 5c) and accordingly, an increase in calcium signals was recorded (Fig. 5d, e). For the single-viral vector strategy, we injected AAV11-hSyn-GCaMP6s vector into the NAc and embedded optical fibers over the VTA (Fig. 5f). In accordance with dual-viral strategy, an increase in calcium signals was also detected after reward stimulation (Fig. 5g). These results showed that AAV11 can be used for functional circuit interrogation.

### AAV11 combined with AAV1 for analyzing neuron-astrocyte connection

Astrocytes are the most abundant glia cells in the CNS. Astrocytes maintain homeostasis of neuronal populations and are intimately involved in various neurological disorders[36,37]. AAV serotypes that can

efficiently target astrocytes are of great interest to the academia and industry. AAV8 and AAV5 are widely used serotypes with astrocyte tropism[38,39]. To evaluate the transduction efficiency of AAV11 on astrocytes, we compared the number of labeled astrocytes after infusion of AAV11 and AAV8 into the right and left sides of the dHPC, respectively (Fig. 6a, b). Both viral vectors carry an EGFP reporter under the control of the astrocyte-specific truncated GfaABC₁D promoter. To evaluate the cell-type specificity of the AAV11-GfaABC₁D, we measured the colocalization of EGFP and astrocyte marker GFAP (Fig. 6e). The results demonstrated that the AAV11-GfaABC₁D driven EGFP expression was mainly distributed in the astroglia (Fig. 6b, e). Remarkably, 21 days post-injection, the EGFP- and GFAP-double positive cell counts were significantly higher on the AAV11 side than on the AAV8 side (Fig. 6f,

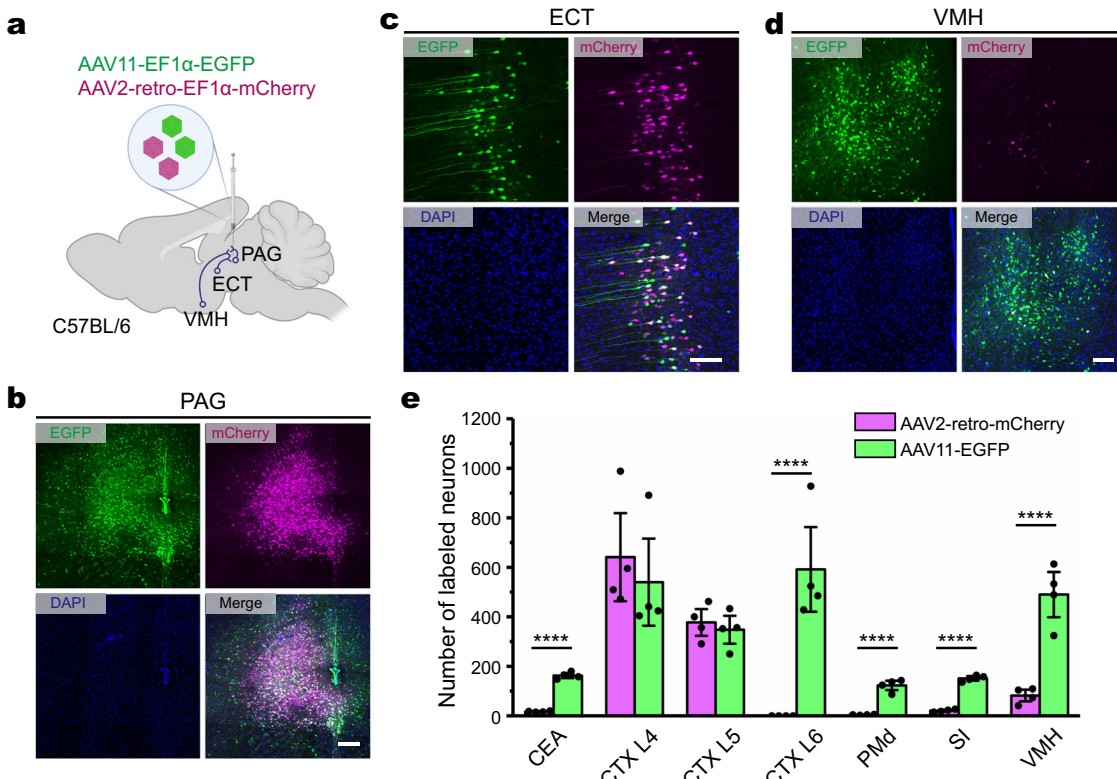

**Fig. 3 | Comparison of the retrograde transduction tropism of AAV11 and AAV2-retro after PAG injection. a** Connectivity diagram shows the injection site (peri-aqueductal gray, PAG) and its upstream areas in ectorhinal area (ECT) and ventromedial hypothalamic nucleus (VMH). AAV11-EF1α-EGFP and AAV2-retro-EF1α-mCherry viruses were mixed at a particle ratio of 1:1 ($1.8 \times 10^9$ VG in total) and injected into PAG of C57BL/6 mice. Diagram was created with BioRender.com. **b** Fluorescence distribution of EGFP (AAV11) and mCherry (AAV2-retro) at the PAG injection site. Scale bar = 200 μm. **c** EGFP (AAV11) and mCherry (AAV2-retro) expression in ECT. Scale bar = 100 μm. **d** EGFP (AAV11) and mCherry (AAV2-retro) expression in VMH. Scale bar = 100 μm. **e** Quantification of AAV11- and AAV2-retro-infected projection neurons. Statistical values are presented as mean ± SEM ($n = 4$/group). Statistical analyses were performed through unpaired two-tailed Student's $t$ tests, with significant differences being expressed by the $p$ value. ****$p < 0.0001$. Source data are provided as a Source Data file.

$p = 0.0017$), implying that AAV11 has a higher potency in targeting astrocytes than AAV8. We also compared the transduction efficiency of AAV11 and AAV5 on astrocytes in the CPu and dHPC (Fig. 6c, d, g, Supplementary Fig. 10). Quantitative analysis showed that AAV11 was more efficient than AAV5 in targeting astrocytes (Fig. 6g, Supplementary Fig. 10c). Similarly, AAV11 infusion also resulted in extensive expression of EGFP in dentate gyrus (DG) astrocytes (Supplementary Fig. 11). Taken together, our study demonstrated that AAV11 is an effective serotype exhibiting considerable astrocyte tropism in vivo.

AAV1 based anterograde axoastrocytic transfer can be used to study neuron-astrocyte connection[40]. Given AAV11 can transduce projection neurons and astrocytes with high efficiency, we hypothesized that AAV11 can be used together with AAV1 to analyze neuron-astrocyte connection. To test this hypothesis, the "leaking" of recombinase-dependent vectors was evaluated (Supplementary Fig. 12), and then we coinjected AAV1-CMV-Cre and AAV5-EF1α-fDIO-tdTomato into the ventral posterior medial nucleus (VPM), coinjected AAV11-GfaABC1D-DIO-EGFP and AAV11-EF1α-DIO-Flp into the barrel cortex (BX) (Fig. 6h). Representative images demonstrated that only VPM neurons projected to BX were marked by tdTomato (magenta), and BX astrocytes connected with VPM neurons were marked by EGFP (green) (Fig. 6i). Therefore, AAV11 can be used in combination with AAV1 to analyze neuron-astrocyte connection.

### Decreased circuit connections in Alzheimer's disease mouse brains revealed by AAV11 tracing

Analyzing the differences of the brain neural networks connection between the healthy and diseased animals is the basis of understanding the mechanism of brain diseases caused by neural network abnormalities[6]. To evaluate whether AAV11 can be used for dissecting the variation of circuit connections in Alzheimer's disease mouse brains, AAV11 was injected into the dHPC area of APP/PS1 (AD) transgenic mice or wild-type (WT) mice (C57BL/6) (Fig. 7a). Three weeks after injection, AAV11 could label neurons at the injection site and the upstream brain regions (Fig. 7b, c). Through quantitative analysis, we found that significantly fewer projection neurons in upstream areas, including the contralateral dHPC (Cont-dHPC), ENT and MSC, were labeled by AAV11 in APP/PS1 (AD) than those in the wild-type C57BL/6 mice (dHPC: $683.33 \pm 73.11$ vs $1098.67 \pm 22.56$, $p = 0.0056$; ENT: $201.00 \pm 40.77$ vs $350.67 \pm 29.81$, $p = 0.0414$; MSC: $17.33 \pm 5.04$ vs $54.33 \pm 10.27$, $p = 0.0319$) (Fig. 7d). The decreased labeling of upstream neurons by AAV11 in APP/PS1 mice suggests a potential reduction in neural circuit connectivity. Alternatively, these results may indicate altered immune responses to AAV or decreased retrograde transport efficiency in 10-month-old APP/PS1 mouse models. This study highlighted the potential utility of AAV11 in circuits related to brain disease.

### Discussion

Recombinant adeno-associated viruses (rAAVs) are considered the safest and most flexible vehicles for DNA delivery, enabling efficacious gene overexpression, knockdown, and editing[41]. AAVs have been applied extensively for tracing and manipulating neural circuits[42], in vivo imaging[43], developing disease models[44], and evaluating molecule-based therapeutic strategies for the treatment of neurological diseases[45]. Routes of CNS-directed AAV infusion include in situ

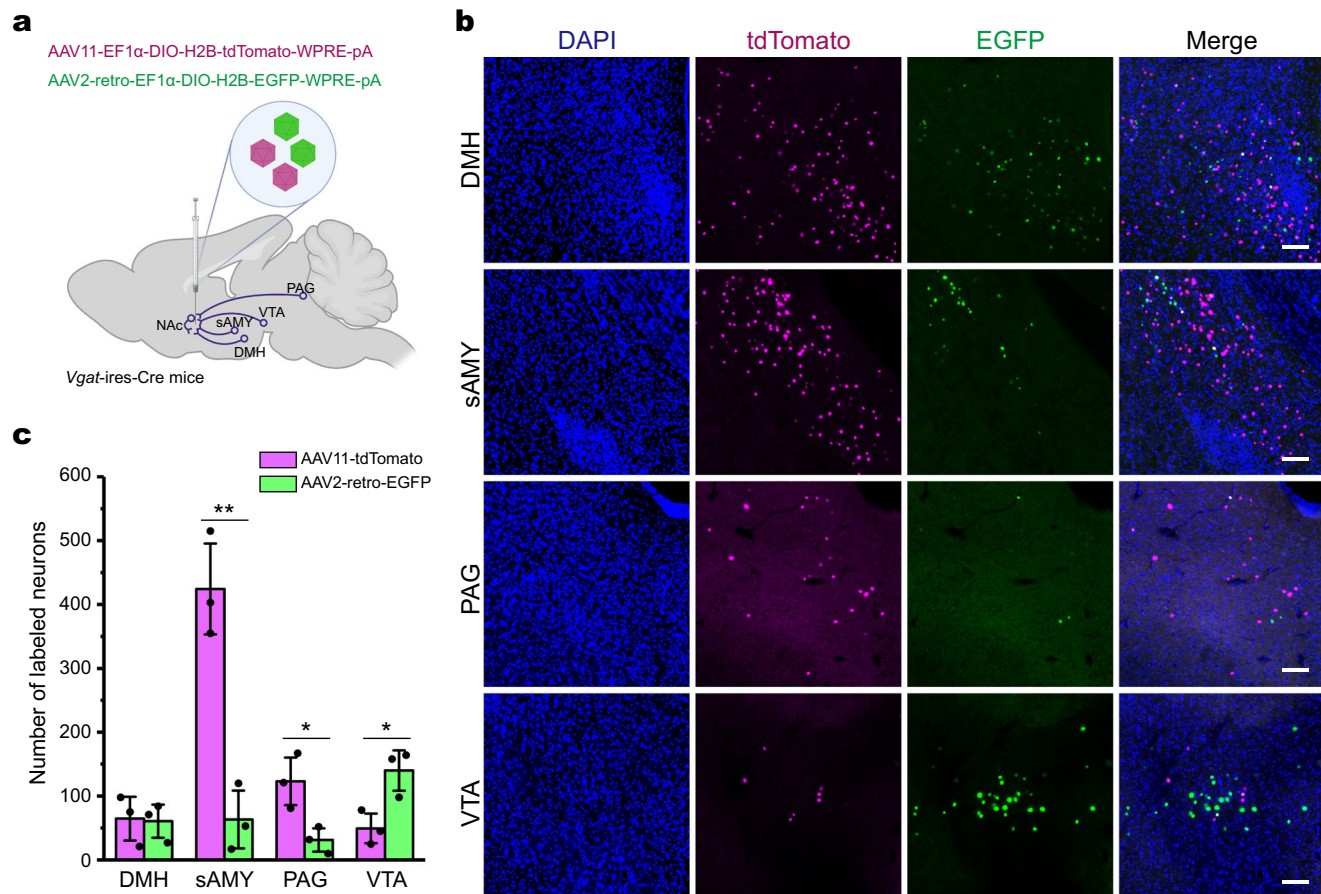

**Fig. 4 | Comparison of AAV11 and AAV2-retro in retrograde targeting of genetically defined neuronal populations. a** Connectivity diagram shows the injection site (nucleus accumbens, NAc) and its upstream areas in dorsomedial nucleus of the hypothalamus (DMH), periaqueductal gray (PAG), striatum-like amygdalar nuclei (sAMY) and ventral tegmental area (VTA). AAV11-EF1α-DIO-H2B-tdTomato and AAV2-retro-EF1α-DIO-H2B-EGFP viruses were mixed at a particle ratio of 1:1 ($3 \times 10^9$ VG in total, 300 nL per mouse) and injected into NAc of *Vgat*-ires-Cre mice. Diagram was created with BioRender.com. **b** Representative images reveal that the retrograde labeling efficiencies of AAV11 and AAV2-retro are quite different in many upstream regions. Scale bar = 100 μm. **c** Quantification of AAV11- and AAV2-retro-infected projection neurons. Statistical values are presented as mean ± SEM (*n* = 3/group). Statistical analyses were performed through unpaired two-tailed Student's *t* tests, with significant differences being expressed by the *p* value. *$p < 0.05$, **$p < 0.01$. Source data are provided as a Source Data file.

injection, blood-brain barrier penetrable intravenous injection[46], anterograde transsynaptic spread[21], and retrograde transport by axon terminal absorption[7]. Here we identified AAV11 as an effective retrograde virus for targeting projection neurons. Compared with AAV2-retro, AAV11 shows comparable or higher potency in retrograde labeling of projection neurons in multiple neural circuits. AAV11 can be readily introduced into the Cre-*lox* system using a dual-viral strategy or combined with a neuron-specific promoter using a single-viral strategy to provide selective and potent gene expression for functional circuit interrogation. Furthermore, in combination with the GfaABC₁D promoter, AAV11 outperforms commonly used AAV8 and AAV5 in transducing astrocytes and can be used together with AAV1 to study neuron-astrocyte connection. Finally, we showed that AAV11 allows for analyzing circuit connectivity difference in the brains of the Alzheimer's disease and control mice. Our study suggested that AAV11 is a powerful tool to investigate complex mechanisms of brain functions and to evaluate the outcomes of gene therapy of neuropsychiatric disorders.

There are various natural serotypes of AAV, which belong to different pedigrees and have their unique tropism[17–19]. Different AAVs possess varying degrees of retrograde labeling properties, while the same serotype may also exhibit neural circuits-dependent targeting efficiencies[7,47]. Screening of retrograde vectors from natural AAV serotypes is feasible and may provide diversified neuroscientific tools for

deciphering brain functions. For example, AAVv66 was recently isolated from the human clinical specimens and demonstrated to share high amino acid sequence similarity with AAV2, but exhibits enhanced CNS transduction compared to AAV2[48]. In this study, we examined the rarely used AAV11, a natural serotype discovered in cynomolgus monkey[49], and surprisingly found that it has retrograde transport capacities superior or complementary to AAV2-retro in some neural circuits. While AAV11 and AAV2-retro shared labeling traits in some brain regions, their unique characteristics provided complementary transduction advantages elsewhere. For example, in the vHPC, AAV2-retro labeled the ENT or PIR retrogradely more effectively, whereas AAV11 showed a greater efficacy in transducing the MSC or EPd (Fig. 2, Supplementary Figs. 5 and 7). Notably, in the CPu, both vectors retrogradely labeled the SSp region, but each demonstrated a preference towards different layers of neuronal populations within the region (Supplementary Fig. 2). To preliminarily investigate the differences in cell types labeled by AAV11 and AAV2-retro, we performed cell type identification of the neuronal population in the SSp area retrogradely labeled from CPu using in situ hybridization and immunofluorescence (Supplementary Fig. 13). While both AAV11 and AAV2-retro labeled mostly excitatory neurons (~90%), AAV11 marked more inhibitory neurons than AAV2-retro in the SSp area (Supplementary Fig. 13c). This efficient transduction of inhibitory neurons by AAV11 was also confirmed in *Vgat*-ires-Cre mice (Fig. 4). However, the tropism of a specific

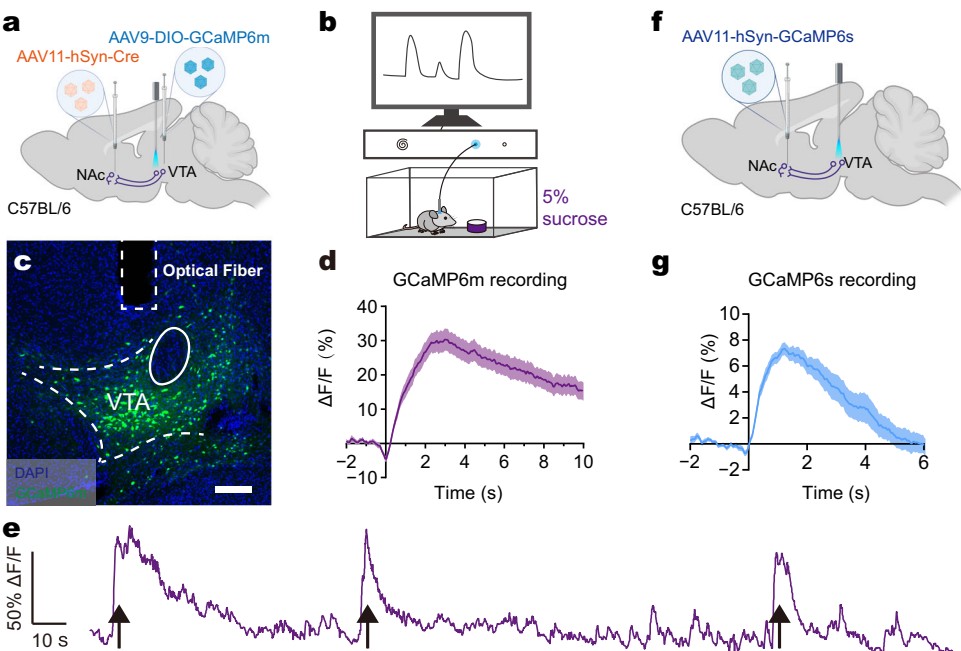

**Fig. 5 | AAV11 can be used for analyzing functional networks. a** Schematic diagram of AAVs injections and fiber implantations to monitor neural activities of NAc-projecting VTA neurons with GCaMP6m. AAV11-hSyn-Cre and AAV9-DIO-GCaMP6m ($2 \times 10^9$ VG each virus) were injected into NAc and VTA, respectively. An optical fiber was implanted into VTA for the detection of calcium flux. Diagram was created with BioRender.com. **b** Schematic diagram of calcium transient monitoring in mice under the reward behavior tasks. **c** The GCaMP6m fluorescence in VTA. Scale bar = 200 μm. **d** Average ΔF/F signals were recorded using dual-viral vector strategy during the behavior tasks. Data were presented as mean ± SEM ($n = 4$/group). **e** A representative trace of GCaMP6m related ΔF/F signals. Black arrows indicate when the animal licked the sucrose solution. **f** Schematic diagram of AAV11-hSyn-GCaMP6s injection and fiber implantations. 100 nL of AAV11-hSyn-GCaMP6s ($2.72 \times 10^{13}$ VG/ml) were injected into NAc. Diagram was created with BioRender.com. **g** Average ΔF/F signals were recorded using single-viral vector strategy during the behavior tasks. Data are presented as mean ± SEM ($n = 4$/group). Source data are provided as a Source Data file.

vector for a certain type of neuron in a particular brain area should not be indiscriminately extended to other brain areas. Detection of the CPu injection site revealed that AAV11 showed greater diffusion than AAV2-retro (Supplementary Fig. 14a), which may contribute to their differential retrograde labeling efficacy by spreading into a larger axon terminal field. Although vHPC could limit the diffusion range of both viral vectors to almost the same extent due to its unique structural characteristics, the transduced cell populations still exhibited differences between the two vectors (Supplementary Fig. 14b). These results imply that the two serotypes recognize different receptors for viral entry and that expression of the receptors varies across brain regions. Most AAVs require both GPR108 and AAVR (KIAA0319L) receptors for cellular entry, except for AAV4 and rh32.33, which are GPR108-dependent yet AAVR-independent[50–52]. Since both AAV11 and rh32.33 belong to AAV4-clade members[18], AAV11 may be dependent on GPR108 for successful cellular transduction. The next important work is to pinpoint the possible binding receptors through single cell sequencing. Meanwhile, this tropism difference hints that the conscious choice of viral tools, either combining two serotypes or using one of them separately, can lead to more satisfying research results in the context of specific neural pathways.

As a vital component of the extensive and intricate neural network, projection neurons are composed of diverse cell types, each with specific cellular and molecular properties, synaptic connections, and in vivo functions[53]. In recent years, many unknown subtypes of projection neurons have been identified through imaging and reconstruction of the complete neuronal morphologies[54,55], but their functions are poorly understood. Recent evidence indicates that GABAergic projection neurons connect many cortical areas unidirectionally or bidirectionally and may participate in the modulation of various behavioral and cognitive functions[56]. On the other hand, abnormal input and output network connection is attributable to

malfunctions of projection neurons. Projection neurons have therefore been considered as therapeutic targets for nervous system diseases[57]. The retrograde viral tools can be adopted for network specific modulation by selectively transducing functional genes into projection neurons between two specific brain regions. In addition, these viral tools can be used in combination with reporters to assess disease-related brain network organization and network reconstruction after treatment (e.g., stem cell therapy)[58,59]. Thus, the AAV11-based viral tools, with their high-level payload expression and stringent retrograde transport properties, will play an important role in further understanding the functions of projection network under different behavior or disease and treatment paradigms.

Another important feature of AAV is astrocyte targeting. AAV5 is a commonly used serotype with astrocyte tropism[38,39]. However, it is well known that AAV serotypes exhibit differential astrocytes targeting efficiency depending on brain region[20]. Diversified astrocyte tropistic viral tools will certainly increase the chance of obtaining optimized research outcomes for a previously untested brain region. Here, we found that AAV11 has higher potency in targeting astrocytes than AAV5, at least in the dorsal hippocampus (dHPC). Our study offers a valuable new option for astrocyte targeting research. Although astrocytes have traditionally been described as supportive partners of neurons, they are now recognized as active participants in the development and plasticity of dendritic spines and synapses[60]. Astrocytic dysfunction has emerged in recent years as a common feature of neurodegenerative disorders such as Alzheimer's disease[61] and Parkinson's disease[62]. Dysregulated astrocytic glutamate uptake, reactivation, and inflammatory responses are frequently demonstrated to drive the pathogenesis of multiple neurodegenerative disorders[60,63]. Astrocytes are therefore considered important therapeutic targets of these diseases. For instance, in vivo astrocytes-neurons conversion is a prospective strategy for the treatment of neurodegenerative

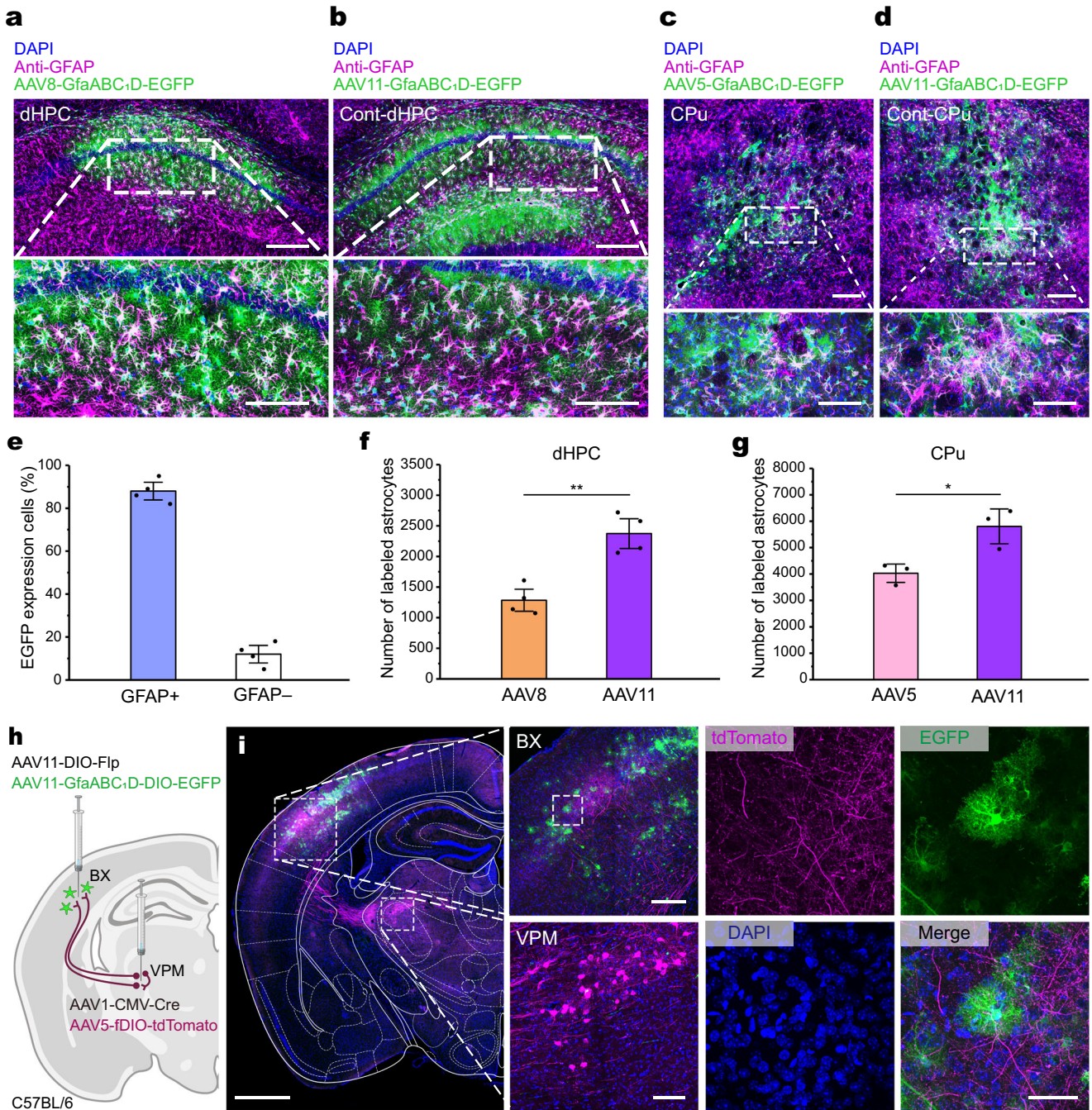

**Fig. 6 | AAV11 combined with AAV1 for analyzing neuron-astrocyte connection.**
Representative images of dHPC astrocytes infected by AAV8 (**a**, left) and AAV11 (**b**, right). 100 nL each of AAV11-GfaABC$_1$D-EGFP and AAV8-GfaABC$_1$D-EGFP ($3 \times 10^9$ VG each virus) were injected into the right and left sides of the dorsal hippocampus (dHPC), respectively. The sections stained with antibody against GFAP (magenta) show the distribution of astrocytes. EGFP signals (green) colocalizing with GFAP staining (magenta) indicate the transduced astrocytes (appeared in white). $n = 4/$ group. Scale bar = 200 μm (top 2 panels), 100 μm (bottom 2 panels). Representative images of caudate-putamen (CPu) astrocytes infected by AAV5 (**c**, left) and AAV11 (**d**, right). 300 nL each of AAV11-GfaABC$_1$D-EGFP and AAV5-GfaABC$_1$D-EGFP ($3 \times 10^9$ VG each virus) were injected into the right and left sides of the CPu, respectively. The sections stained with antibody against GFAP (magenta) show the distribution of astrocytes. EGFP signals (green) colocalizing with GFAP staining (magenta) indicate the transduced astrocytes (appeared in white). $n = 3/$group. Cont-CPu contralateral CPu. Scale bar = 200 μm (top 2 panels), 100 μm (bottom 2

panels). **e** Analysis of the colocalization of EGFP signals and GFAP staining. Statistical values are presented as mean ± SEM ($n = 4/$group). **f, g** Quantification of astrocyte-specific transduction in the indicated regions. Statistical values are presented as mean ± SEM ($n = 4$ in dHPC group and $n = 3$ in CPu group). Statistical analyses were performed through unpaired two tailed Student's $t$ tests, with significant differences being expressed by the $p$ value. $^{**}p < 0.01$, $^{*}p < 0.05$. Source data are provided as a Source Data file. **h** AAVs labeling strategy for analyzing neuron-astrocyte connection: AAV1-CMV-Cre and AAV5-EF1α-fDIO-tdTomato were coinjected into the VPM, and AAV11-GfaABC$_1$D-DIO-EGFP and AAV11-EF1α-DIO-Flp were coinjected into BX. Only VPM neurons projected to BX can be marked by tdTomato (magenta), and BX astrocytes connected with VPM neurons can be marked by EGFP (green). Diagram was created with BioRender.com.
**i** Representative images of Cre-dependent EGFP (green) labeling of BX astrocytes and tdTomato (magenta) labeling of VPM neurons, 3 weeks after AAVs injection.

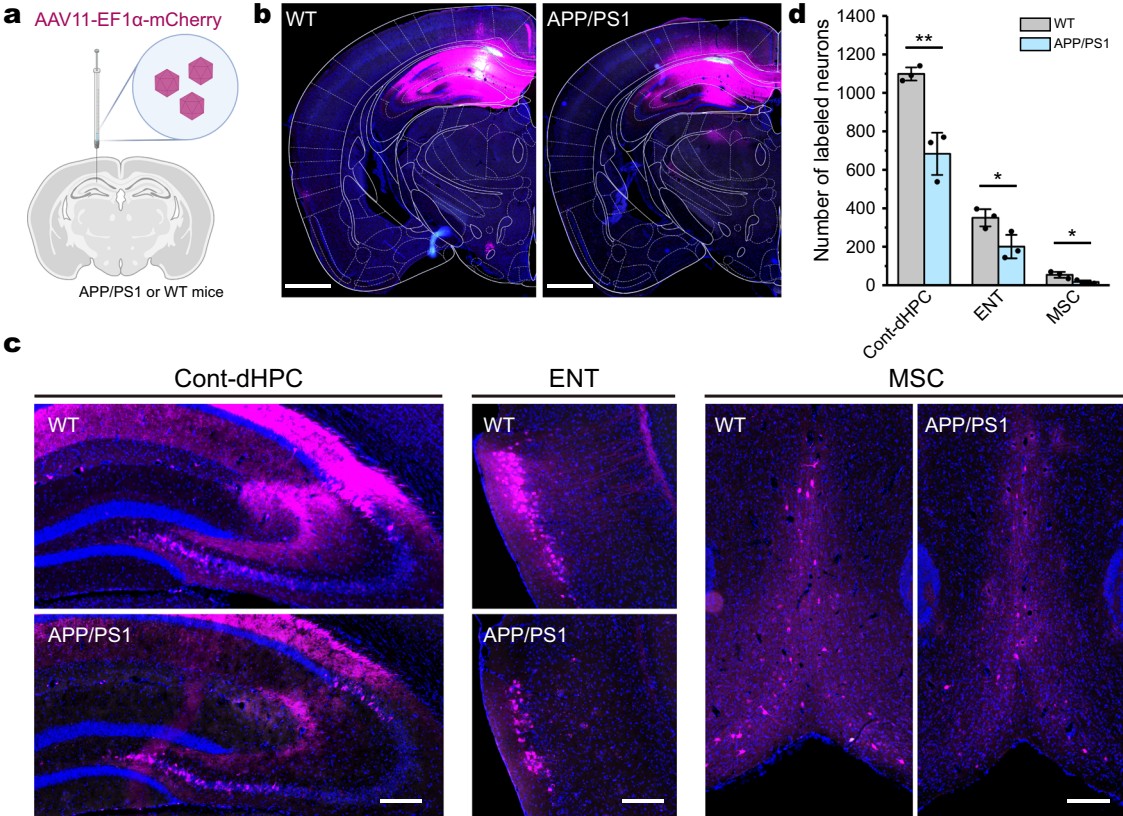

**Fig. 7 | Comparison of circuit connections between Alzheimer's disease and wild-type mice using AAV11 tracing. a** Schematic diagram of virus injection. AAV11-EF1α-mCherry ($6 \times 10^9$ VG per mouse) and CTB-488 (cholera toxin subunit B binding fluorescein 488, used to indicate the injection site; Thermo Fisher Scientific) were mixed (a volume ratio of 5:1, 200 nL per mouse) and injected into the dorsal hippocampus (dHPC) area of APP/PS1 (AD) transgenic mice and wild-type (WT) mice (C57BL/6) respectively. Diagram was created with BioRender.com. **b** The fluorescence distribution of mCherry at the injection site of dHPC. Scale bar = 1 mm. **c** Representative images of labeled neurons in upstream areas of dHPC, 3 weeks after AAVs injection. Scale bar = 200 μm. **d** Quantitative of mCherry-labeled projection neurons in upstream areas. Cont-dHPC contralateral dHPC, ENT entorhinal cortex, MSC medial septal complex. Statistical values are presented as mean ± SEM ($n = 3$/group). Statistical analyses were performed through unpaired two-tailed Student's $t$ tests, with significant differences being expressed by the $p$ value. *$p < 0.05$,**$p < 0.01$. Source data are provided as a Source Data file.

diseases[39]. Downregulation of polypyrimidine tract-binding protein 1 (PTB) using in vivo viral delivery of shPTB or RNA-targeting CasRx resulted in highly effective conversion of astrocytes to neurons and alleviated symptoms of neurodegenerative diseases[64,65]. In this regard, AAV vectors that can efficiently and specifically target astrocytes underpin such a therapeutic strategy. Our study highlighted AAV11 as a powerful and promising viral vector for astrocyte targeting to treat neurodegenerative disorders.

Based on the above transduction tropism of the nervous system, AAV11 can become a new engineering target for virus vector designers. The advancement of the AAV2-retro virus benefits from AAV capsid engineering. There are currently at least four main approaches to capsid development: directed evolution[7,66], rational design[6], in silico evolutionary lineage analysis[67], and natural discovery[17]. We have cloned the naturally occurring AAV11 capsid gene fragment into the packaging plasmid of the triple-plasmid system and therefore provide a readily editable toolkit for AAV11 modification. Referring to the previous experiences in AAV capsid engineering, it is possible to obtain mutant versions of AAV11 through artificial upgrades to improve retrograde labeling or astrocyte targeting ability. For example, we deduced a site on the AAV11 capsid protein suitable for peptide presentation by structural analysis and attempted to present the reported functional peptides (e.g., retro peptide[7] or PHP.eB peptide[66]) on the surface of the capsid protein to further improve the transduction properties of modified AAV11. In addition, exploring the receptors bound by AAV11 or AAV2-retro to infect neurons will also provide a

reference for the upgrade of retrograde vectors. In general, a number of AAV11-based viral tools are expected to be generated soon, which will definitely advance neural network research.

## Methods
### Plasmids construction and virus manufacturing
To obtain the pAAV2/11 plasmid, AAV11 *Cap* sequence was retrieved from NCBI Genbank (accession number: AY631966.1) and synthesized as a template. The AAV11 Cap fragment was amplified using PrimeSTAR HS DNA Polymerase (TaKaRa, Kyoto, Japan, R040A) with primer C11F (5′- ATGATTTAAATCAGGTATGGCT GCTGACGGTTATCTT-3′) and C11R (5′- TCAACCGGTTTATTGA TTAACACGTAATTACAAATGATTAGTCAAATAACGAGAGCCAATAA CC-3′). The amplified product was digested with SwaI and AgeI restriction endonucleases (New England Biolabs, Ipswich, MA, USA) and ligated into the pAAV-RC2/1 vector (Addgene, Watertown, MA, USA, 112862) using T4 DNA ligase (New England Biolabs, Ipswich, MA, USA, M0202M). The recombinant plasmid was then transformed into chemically competent *E. coli* stain Stbl3. The positive clone was picked after PCR identification and the plasmid was extracted to obtain pAAV-RC2/11 packaging plasmid. The transfer plasmids carrying the fragment encoding reporter or functional genes (e.g., pAAV-EF1α-EGFP-WPRE-pA, pAAV-EF1α-mCherry-WPRE-pA, pAAV-EF1α-DIO-H2B-EGFP-WPRE-pA, pAAV-EF1α-DIO-H2B-tdTomato-WPRE-pA, pAAV-EF1α-DIO-GCaMP6m-WPRE-pA, pAAV-hSyn-GCaMP6s-WPRE-pA, pAAV-GfaABC₁D-EGFP-

WPRE-pA, pAAV-CMV-Cre-WPRE-pA, pAAV-GfaABC₁D-DIO-EGFP-WPRE-pA, pAAV-EF1α-DIO-Flp-WPRE-pA, pAAV-EF1α-fDIO-tdTomato-WPRE-pA, pAAV-CAG-EGFP-WPRE-pA or pAAV-hSyn-Cre-WPRE-pA) were co-transfected into HEK-293T cells with the helper plasmid pAdDeltaF6 (Addgene, Watertown, MA, USA, 112867) and different packaging plasmids (e.g., pAAV-RC2/11, pAAV-RC2-retro, pAAV-RC1, pAAV-RC5, pAAV-RC8 or pAAV-RC9) at the molecular ratio of 1:1:1. Each AAV was prepared using 15 cell culture dishes (100 × 20 mm). Viral particles were harvested at 72 hours post-transfection and purified by iodixanol gradient ultracentrifugation[68]. The purified rAAVs were titered by qPCR using the iQ SYBR Green Supermix kit (Bio-Rad, Hercules, CA, USA, 1708884). All viral vectors were aliquoted and stored at −80 °C until use.

## Research animals

8–10 week-old adult male C57BL/6 mice (Hunan SJA Laboratory Animal Company, Changsha, Hunan, China), Ai14 transgenic mice (The Jackson Laboratory, Bar Harbor, ME, USA) and *Vgat*-ires-Cre transgenic mice (The Jackson Laboratory, Bar Harbor, ME, USA) were used for experiments. Additionally, ten-month-old APP/PS1 (AD) transgenic mice and wild-type (WT) C57BL/6 mice (provided by the Xuzhou Medical University, Xuzhou, Jiangsu, China) were used for detecting circuit changes in Alzheimer's disease model mice. The mice were housed under a 12/12-h light/dark cycle in specific pathogen-free facilities with controlled temperature (22–24 °C) and humidity (40–60%), water and food were supplied *ad libitum*. All the surgical and experimental procedures were performed following the guidelines formulated by the Animal Care and Use Committee of Innovation Academy for Precision Measurement Science and Technology, Chinese Academy of Sciences.

## Stereotaxic AAV injection

Mice were deeply anaesthetized using 1% pentobarbital intraperitoneally (i.p., 50 mg/kg body weight). The stereotactic injection coordinates were selected according to Paxinos and Franklin's *The Mouse Brain in Stereotaxic Coordinates*, 4th edition[69]. Animals were placed on a stereotactic frame (RWD, Shenzhen, Guangdong, China, 68030). A small volume of virus was injected into the CPu (relative to bregma: anterior-posterior-axis (AP) + 0.80 mm, medial-lateral-axis (ML) ± 2.00 mm, and dorsal-ventral-axis (DV) −3.30 mm), vHPC (relative to bregma: AP −3.16 mm, ML ± 2.95 mm, and DV −4.10 mm), PAG (relative to bregma: AP −4.00 mm, ML ± 0.26 mm, and DV −2.60 mm), NAc (relative to bregma: AP + 1.50 mm, ML ± 1.10 mm, and DV −4.60 mm), VTA (relative to bregma: AP −3.20 mm, ML ± 0.45 mm, and DV −4.30 mm), dHPC (relative to bregma: AP −2.00 mm, ML ± 1.40 mm, and DV −1.50 mm), VPM (relative to bregma: AP −1.80 mm, ML ± 1.7 mm, and DV −3.50 mm), BX(relative to bregma: AP −1.80 mm, ML ± 3.2 mm, and DV −0.50 mm), V1 (relative to bregma: AP −3.90 mm, ML ± 2.6 mm, and DV −1.30 mm) and DG (relative to bregma: AP −2.15 mm, ML ± 1.30 mm, and DV −2.00 mm), at a rate of 0.03 µL/min using a stereotaxic injector equipped with a pulled glass capillary (Stoelting, Wood Dale, IL, USA, 53311). After the injection was complete, the micropipette was held for an additional 10 min before being withdrawn. Animals were allowed to recover from anesthesia on a heating pad. Three weeks after injection, animals were anesthetized with 1% sodium pentobarbital, administered intraperitoneally. Secured on a foam board, they then underwent cardiac perfusion. Upon completion, brain tissue was collected for analysis. After overnight post-fixation in 4% paraformaldehyde solution, brains were dehydrated in 30% sucrose solution for one day.

## Slice preparation, immunofluorescence, and imaging

Slice preparation and imaging were completed according to the previously reported methods[6]. Coronal sections (40 µm) were cut on a microtome (Thermo Fisher Scientific, Waltham, MA, USA), collected in anti-freeze fluid, and stored at −20 °C for further use. For GFAP staining, sections were incubated with goat anti-GFAP (1:800, Abcam, Cambridge, MA, USA, ab53554) primary antibody and then with rabbit anti-goat IgG conjugated with Cy3 (1:400, The Jackson Laboratory, Bar Harbor, ME, USA, 305-165-003) secondary antibody. For CaMKIIα staining, sections were incubated with rabbit anti-CaMKIIα (1:400, Abcam, Cambridge, MA, USA, ab5683) primary antibody and then with Goat Anti-Rabbit IgG H&L Alexa Fluor 647 (1:400, The Jackson Laboratory, Bar Harbor, ME, USA, 111-605-003) secondary antibody. For GABA staining, sections were incubated with mouse anti-GABA (1:400, Sigma-Aldrich, St. Louis, MO, USA, A0310) primary antibody and then with Donkey anti-Mouse Alexa Fluor 647 (H&L) (1:400, The Jackson Laboratory, Bar Harbor, ME, USA, 715-605-151) secondary antibody. After thorough washing with PBS, sections were counterstained with DAPI (1:4000, Beyotime, Shanghai, China) and mounted using 70% glycerol. Imaging was performed using the Olympus VS120 Slide Scanner microscope (Olympus, Tokyo, Japan) or Leica TCS SP8 confocal microscope (Leica, Wetzlar, Germany).

## Fluorescent in situ hybridization

Spatial FISH Ltd (Shenzhen, Guangdong, China) designed the specific probes for target RNA. Samples were fixed with 4% paraformaldehyde and covered with a reaction chamber for the following reactions. After dehydration and denaturation with methanol, the hybridization buffer containing specific targeting probes was added to the chamber and incubated at 37 °C overnight. Samples were then washed three times with PBST before the target probes were ligated in ligation mix at 25 °C for 3 hours. Following this, samples were washed three times with PBST and subjected to rolling circle amplification by Phi29 DNA polymerase at 30 °C overnight. The fluorescent detection probes in hybridization buffer were then applied to samples. Finally, samples were dehydrated with an ethanol series and mounted with mounting medium. After capturing images, signal dots were decoded to interpret RNA spatial position information.

## GCaMP6-based calcium imaging in vivo

AAV11-hSyn-Cre and AAV9-DIO-GCaMP6m was injected into the NAc and VTA, respectively. AAV11-hSyn-GCaMP6s was injected into the VTA. Optical fiber (core diameter: 200 µm, numerical aperture: 0.37, Inper, Hangzhou, Zhejiang, China) was implanted into the VTA. Mice had visually identifiable GCaMP6-expressing cells in VTA two weeks after injection. Before recording, the mice were handled for 3–5 min for at least 3 days and then habituated to the fiber patch cord and the chamber (20 × 20 × 22 cm) for 10 min. All mice were deprived of water supply for 24 h and placed in the chamber equipped with a cup filled with 5% (w/v) sucrose solution. Calcium transients were recorded by exciting GCaMP6 at 470 nm using the fiber photometry system (ThinkerTech, Nanjing, Jiangsu, China).

## Statistics and reproducibility

Data were analyzed using GraphPad Prism 7.0 (GraphPad Software, La Jolla, CA, USA), Origin 7.0 (OriginLab, Northampton, MA, USA), MATLAB R2018b (MathWorks, Natick, MA, USA) and Microsoft Excel (Microsoft, Redmond, WA, USA). For cell counting, one-sixth of brain slices from each animal were selected. Positive cells were quantified using ImageJ software v1.8.0 (National Institutes of Health, Bethesda, MD, USA). Data were shown as means ± standard error of the mean (SEM). All statistical analyses were performed through unpaired two-tailed Student's *t* tests, with significant differences being expressed by the *p* value. All experiments were repeated at least three times, with all repetitions yielding consistent results.

## Reporting summary

Further information on research design is available in the Nature Portfolio Reporting Summary linked to this article.

## Data availability

The authors declare that all supporting data for this study are included within the manuscript and its Supplementary Information files or can be obtained from the authors. The nucleic acid sequence for AAV11 is accessible through the NCBI GenBank (accession number: AY631966.1). The relevant source data, including those for Figs. 2–7 and Supplementary Figs. 1, 3, 10, 13, are consolidated in the Source Data file accompanying this paper. Source data are provided with this paper.

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

## Acknowledgements

We are grateful to Liting Luo and Lingling Xu (Core Facility Center, Innovation Academy for Precision Measurement Science and Technology, Chinese Academy of Sciences) for technical assistance with centrifugation and microscope imaging. We appreciate Dr. Yingwei Zheng (Xuzhou Medical University) for providing the APP/PS1 transgenic mice. This work was supported by the STI2030-Major Projects (2021ZD0201003), the National Natural Science Foundation of China (31830035, 31771156, 21921004, 82151309), the Key-Area Research and Development Program of Guangdong Province (2018B030331001), the Strategic Priority Research Program of the Chinese Academy of Sciences (XDB32030200) and the Shenzhen Key Laboratory of Viral Vectors for Biomedicine (ZDSYS20200811142401005), and the Key Laboratory of Quality Control Technology for Virus-Based Therapeutics, Guang-dong Provincial Medical Products Administration (2022ZDZ13).

## Author contributions

KL and FX contributed to the study idea and design; FX, JW and CY contributed to funding acquisition and resources; ZH, NL, WM, YC, JK, LL, SP, ZX, WZ and YQ performed the experiments and data acquisition; ZH, NL, KL and YW accomplished data analysis; ZH, KL, XL and FX drafted the manuscript, and contributed to review and editing. All authors read and approved the final manuscript.

## Competing interests

FX is a scientific co-founder of BrainCase Biotechnology Co., Ltd and holds equity in the company. KL, ZH, NL and FX are inventors on two patents derived from this manuscript, which have been submitted to the Patent Office of the People's Republic of China (Application Nos. 202111488120.7 and 202111486070.9) by the Shenzhen Institute of Advanced Technology, Chinese Academy of Sciences. The remaining authors declare no competing interests.

## Additional information

[1]Shenzhen Key Laboratory of Viral Vectors for Biomedicine, Shenzhen-Hong Kong Institute of Brain Science, Shenzhen Institute of Advanced Technology, Chinese Academy of Sciences, Shenzhen 518055, PR China. [2]Key Laboratory of Magnetic Resonance in Biological Systems, State Key Laboratory of Magnetic Resonance and Atomic and Molecular Physics, National Center for Magnetic Resonance in Wuhan, Innovation Academy for Precision Measurement Science and Technology, Chinese Academy of Sciences, Wuhan 430071, PR China. [3]Key Laboratory of Quality Control Technology for Virus-Based Therapeutics, Guangdong Provincial Medical Products Administration, NMPA Key Laboratory for Research and Evaluation of Viral Vector Technology in Cell and Gene Therapy Medicinal Products, the Brain Cognition and Brain Disease Institute, Shenzhen Institute of Advanced Technology, Chinese Academy of Sciences, Shenzhen 518055, PR China. [4]University of Chinese Academy of Sciences, 100049 Beijing, PR China. [5]Wuhan National Laboratory for Optoelectronics, Huazhong University of Science and Technology, Wuhan 430074, PR China. [6]Department of Anaesthesia and Intensive Care, Peter Hung Pain Research Institute, The Chinese University of Hong Kong, Hong Kong SAR, PR China. [7]Department of Pathophysiology, School of Basic Medicine, Tongji Medical College, Huazhong University of Science and Technology, Wuhan 430030, PR China. [8]College of Life Sciences, Wuhan University, Wuhan 430072, PR China. [9]Center for Excellence in Brain Science and Intelligence Technology, Chinese Academy of Sciences, Shanghai 200031, PR China. [10]These authors contributed equally: Zengpeng Han, Nengsong Luo. ✉e-mail: kz.lin@siat.ac.cn; fq.xu@siat.ac.cn

