## [Peer Review File · Nature Communications]

AAV11 enables efficient retrograde targeting of projection neurons and enhances astrocyte-directed transductionREVIEWER COMMENTS

Reviewer #1 (Remarks to the Author):

The authors present a very interesting study "AAV11 performs efficient retrograde targeting of projection neurons and enhances astrocyte-directed transduction" and present evidence to suggest this has enhanced efficiency over other AAV vector systems for retrogradely labeling brain regions. They also demonstrate that it can effectively deliver calcium indicators for monitoring functional networks and it has a superior tropism for astrocytes than commonly used serotypes.

The manuscript is well written, the figures are clear and the methods are detailed.

However, there are a number of issues regarding the data that need to be addressed. In Suppl fig 2 the authors show a comparison between AAV11 and AAV2-retro on retrograde transduction there appears to be big differences in the populations transduced with a significant number of cells being transduced only with AAV2 or only AAV11. Would you not expect greater colocalization of the vectors? Is this an artifact of the technique where expression from one vector effects expression from the other in a cell or are they potentially targeting 2 different populations? This is an important consideration if this approach is to be used to efficiently retrogradely transduce a brain region.

For the data on astrocyte transduction there does appear to be a greater number of GFAP positive cells transduced for AAV11 there are cells that are transduced that are not GFAP positive. If the aim is to target astrocytes for gene delivery in disease, specificity is also likely to be crucial. The authors need to present data on what percentage of transduced cells are GFAP positive, what percentage of GFAP positive cells are transduced, and what the non-GFAP labeled cells are transduced.

While this paper shows some compelling data regarding the ability to retrogradely label brain areas it would have been nice to see some data showing how this could be used to monitor changes due to disease. For example, modifying the functional networks to show a change in calcium signaling or changes in connectivity due to damage to a specific brain area.

Reviewer #2 (Remarks to the Author):

Summary

Retrograde transduction is a feature of many viral vectors, particularly AAVs. Neuroscientists leverage retrograde transduction to study circuit connectivity and to deliver sensors, reporters, and effectors (opsins and DREADDs) to populations of cells that send projections to specific brain regions. AAVs are particularly useful for these studies because they can mediate long-term, strong expression with less neurotoxicity than many other viral vectors. Previously, researchers developed AAV2-Retro, a modified AAV capsid that has improved retrograde transport properties when injected into the rodent or primate brain. This capsid has become commonly used (cited over 700 times) due to the need for efficient

retrograde vectors. Here, Han et al. demonstrate the novel finding that a less well studied natural AAV serotype, AAV11 has significant retrograde transduction properties in the mouse brain. The authors compare AAV11 and AAV2-Retro head-to-head in several interconnected brain regions and demonstrate that AAV11 outperforms AAV2-Retro in the transduction of several projection neuron populations (e.g., the CEA and L6 CTX cells after injection into the PAG). They show retrograde transduction with AAV11 paired with Vgat-Cre transgenic mice to label GABAergic projection neurons that send projections to the NAc and they demonstrate, using fiber photometry experiments, that expression from AAV11 is sufficient to detect neuronal activity in VTA neurons projecting to the NAc. Finally, they also demonstrate that AAV11, like many AAVs, transduces astrocytes, a finding that is incomplete and off-topic for this otherwise focused study.

Major concerns

The authors have not commented on the possible increased local spread of AAV11 vs AAV2-Retro at the injection site (which can be seen Fig 4 and Supplementary Fig 2). Can the increased transduction observed in select neuronal populations be related to the fact that those populations innervate regions surrounding the injection site? The extent of local spread is an important consideration for retrograde labeling experiments, and there are use cases for vectors with wide or limited spread.

Additional characterization of the extent of local spread is an important part of the characterization of AAV11 relative to AAV2-Retro that is lacking in this study.

The abstract highlights AAV11 as a capsid that “has greatly enhanced efficiency over AAV2-Retro”. While true in several specific cases, this appears to be an overstatement. The authors found that AAV2-Retro also outperformed AAV11 in several populations. This manuscript could have the most impact on the field if it gave researchers a thorough and unbiased quantitative comparison between AAV11 and AAV2-Retro.

Specific concerns

The authors repeatedly reference the potential therapeutic use cases for circuit manipulation gene therapy without citing specific examples. It seems premature to be promoting this as a current use case for AAV11 or other vectors.

The authors cite ref 26 in a statement about rational design in the introduction about retrograde vectors. This is an inappropriate citation: Ref 26 is a study on IV administered vectors unrelated to retrograde transport. This work is also cited later for rational design, but this study used panning of a phage display library to identify peptides that were then transferred to the AAV capsid, therefore it is not an example of rational design.

The authors refer to different types of AAVs as pedigrees. Clade or serotype would be a more appropriate description depending on whether the authors seek to distinguish between AAVs based on sequence similarity or differential resistance to antibodies.

At the end of the introduction, the authors speculate about the potential use of AAV11 as a base capsid

that could be further engineered to improve its properties. This text should be removed from the introduction. This is an inappropriate place for this, as it will confuse the reader about what was actually done in the study.

The entire first paragraph of the results section describes the author's choice not to use the baculovirus system for AAV11 production and to move the AAV11 cap gene into a plasmid for triple transfection. As the later is the standard approach for making research scale AAV, this section seems unnecessary and is better described in the methods, if at all. The only important statement is in the final sentence of the paragraph where the authors describe production yield data.

One well studied population that AAV2-Retro has been reported to target poorly are dopaminergic neurons in the SNc. Given that the authors injected AAV2-Retro and AAV11 into the CPU, they should include a comparison of the transduction of this population or alternatively assess whether their NAC injections which provided expression in the VTA work better with AAV11 than AAV2-Retro.

Previously the authors reported AAV9-Retro, a capsid they generated by transferring the AAV2-Retro insertion to the same region of the AAV9 capsid. Have the authors compared AAV11 with AAV9-Retro?

In the section on anterograde transport, the final conclusion is overstated, "These results indicate that AAV11 doesn't spread anterograde". The authors should qualify this statement by saying it does not provide anterograde transduction in the tested circuit.

In the results, the authors cite a paper (reference 38) for the use of AAV2-Retro for therapeutic evaluations. However, this paper used AAV2-Retro to express a partial Huntingtin gene fragment (modeling HD, rather than testing a potential therapeutic).

The astrocyte studies are weak and distract from the primary finding of the paper. If the authors wish to include these findings, it would be important to investigate this finding in more depth. Is this finding true across multiple brain regions? The authors cite ref 30, stating that there is variability in the transduction of astrocytes across brain regions by different AAV vectors. Notably, in this prior study, AAV5, which the authors compare with AAV11, underperforms AAV8 in all brain regions, and particularly in the hippocampus (the location the authors chose to examine in the current study). This calls into question the importance and generalizability of their finding. How does AAV11 compare with the more potent AAV8 vector? Does AAV11 provide more efficient transduction of astrocytes in the striatum, a region where AAV5 was more effective? Furthermore, in Fig 7 they inappropriately use $n=27$ (number of sections examined) rather than an $n = 3$ (number of mice examined). As injection site difference can be a major source of variability, this inflates the n and possibly the apparent statistical significance of the findings. As astrocyte transduction is not the primary focus of the paper, the authors could alternatively consider describing the astrocyte transduction (so future users are aware) without making quantitative claims.

Supplementary Fig 4 has the title "AAV11 for specifically targeting astrocytes". However, the authors do

not show any data or quantitative results demonstrating that AAV11 is specifically targeting astrocytes, nor is this consistent with their other retrograde data. AAV11 is transducing neurons and glia, but they are using a GfABC1D promoter to limit expression to astrocytes. The authors also do not present any quantitative data showing what fraction of the GFP+ cells are astrocytes.

Given that the true variable the authors are seeking to study is AAV11 vs AAV2-Retro, and given that virus preps can vary significantly from lot to lot, it is imperative that the authors comment on how many independently produced AAV11 and AAV2-Retro lots were compared in this study. If all of these comparisons were from a single lot of either vector, the authors should evaluate and report results from additional lots of both vectors in at least a subset of these assays to ensure reproducibility of their findings.

The authors claim in their abstract and discussion that they showed the potential of AAV11 to monitor activities in a functional network using a cell-type-specific promoter, but used hSyn as the promoter for this study. This promoter is broadly expressed in nearly all neuron populations, so this claim is overstated.

The authors have not provided any description of their cell counting methods or the statistical tests used. The statistical tests used should also be described in each relevant figure legend.

Reviewer #3 (Remarks to the Author):

This manuscript by Han et al reports the use of naturally occurring AAV11 serotype for retrograde labeling of neurons, which allows for neuronal circuit interrogation in the CNS. Overall, this is an interesting finding, however, the finding is only incremental in nature when one considers that AAV2-retro has already been developed for retrograde labeling of circuits. Many of the findings reported in this manuscript suggest that AAV11 is an optimized version of AAV2-retro, with the ability to detect a larger number of retrogradely labeled neurons or the ability to label a few difficult to label circuits in the brain. In addition to the points above that relate to the impact and significance of this study, there are major points and other points that will need to be considered by the authors for the manuscript. These are listed below:

MAJOR POINTS:

(1) The data on astrocytes (Fig. 7) appear to be completely out of context with the manuscript that has the goal of examining AAV11 as a retrograde reporter virus. In addition, data shown in Fig 7d suggest a very modest increase in the ability of AAV11 to label astrocytes compared to the well used and well characterized AAV5 serotype. The impact of this manuscript will be greatly increased if the authors can show that AAV11 can label axonal processes within the territory of transduced astrocytes, which would be a very useful and impactful tool for the field. In fact a recent study (PMID: 35138891) has shown

axoastrocytic AAV transfer using novel methods. The authors could use similar methods to examine if AAV11 is capable of astroaxonal transfer.

(2) All figures show very low resolution / magnification images of cell bodies. It is important to show zoomed in confocal images of the cell bodies and quantify cell bodies labeled for AAV2-retro and AAV11 in this way. This is especially important because the quantal yield of mcherry is several fold lower than GFP, which can easily lead to a wrong interpretation that the labeling efficiency of AAV2-retro is lower than that of AAV11. This fact holds true for all the data shown in this manuscript.

(3) It is unclear how the use of AAV11 as shown in Figure 6 provides advantages over existing methods to label neurons with GCaMPs. For instance it is easily possible to directly label the neuronal nucleus of interest using the multiple Cre mouse lines. The authors need to design and perform these experiments in a way that truly bring out the advantage of using AAV11 over AAV2-retro in the context of interrogating functional circuits.

OTHER POINTS:

(1) Title: The use of the word "performs" in the title is odd. Please revise the title.

(2) Line 48: The word "mental" should be "neurological"

(3) Line 60: Please add references for each of the viral vector types mentioned in this sentence

(4) Supplementary Fig 1: Titers are quite low. The methods do not details if and how a large scale AAV prep was made. Did the authors use a 10-layer cell culture system, and how was the vector prep purified. These details need to be mentioned in the methods.

(5) Line 107: The word "infectious effect" is ambiguous and odd. Please rephrase this line.

(6) Fig 1a and Fig 2 need to show labeling of the SNc and VTA, which are major nuclei projecting to the NAc for both retro-AAV2 and AAV11. Confocal microscopy with high resolution imaging of the SNc and VTA would be necessary here.

(7) Supplementary Fig 3: One important control here is to use AAV2-mcherry (non-retro version) as a negative control. Sections that are 2.2, 0.5, -0.5 mm from bregma look different in many regions. This experiment needs to be repeated with dual color labeling in the same animal (AAV2-retro-mcherry + AAV11 GFP).

(8) Line 126: "doesn't" needs to be "does not"

(9) Fig 5c needs to show a head-to-head comparison with AAV2-retro-mcherry virus using the DIO flex system in AAV2-retro

(10) Line 173: "GCaMP detects calcium flux at the scale of single action potentials". This sentence is wrong. GCaMPs do not have the resolution to detect single action potential calcium fluxes. The data reported by the authors are very likely to be because of burst firing of neurons. Please correct this sentence.

(11) Discussion: There needs to be more discussion on the mechanism, receptors etc that may be involved in retrograde infection of axonal terminals.

(12) Methods: The methods section lacks a description of how the neuronal cell bodies were quantified in this study. Did the authors use ImageJ or some other program? Were the cell bodies thresholded for analysis?

Responses to reviewers' comments

Point-to-point responses to reviewer's comments (marked by blue):

The new experimental results are briefed as following:

1. More finely characterize the retrograde labeling efficiency of AAV11 compared to AAV2retro as following:
 - (1) Reducing the injection dosage of AAV11 (**Supplementary Fig. 2 and Supplementary Fig. 8**)
 - (2) Separate injection (**Supplementary Fig. 5**)
 - (3) Different virus batches (**Supplementary Fig. 6**)
 - (4) Exchange of the carried fluorescent gene (**Supplementary Fig. 7**)
2. More comprehensive characterization of AAV11 transduction of astrocytes. compared the transduction efficiency of AAV11 and AAV8 in dHPC (**Fig.6 a-d**). compared the transduction efficiency of AAV11 and AAV5 on astrocytes in CPu and dHPC (**Supplementary Fig. 10**).
3. Utilize the ability of AAV11 retrograde tracing with astrocytes transduction to analyze the neural-astrocyte circuit connection (**Fig.6 e and f**).
4. Head to head comparison the efficiency between AAV11 and AAV2-retro in retrograde transduction of GABAergic neurons projecting to NAc. (**Fig.4**)
5. Use AAV11 to dissect the variation of circuit connections in Alzheimer's disease mouse brains (**Fig.7**).
6. Imaging was performed using the Leica TCS SP8 confocal microscope.

Reviewer #1 (Remarks to the Author):

The authors present a very interesting study "AAV11 performs efficient retrograde targeting of projection neurons and enhances astrocyte-directed transduction" and present evidence to suggest this has enhanced efficiency over other AAV vector systems for retrogradely labeling brain regions. They also demonstrate that it can effectively deliver calcium indicators for monitoring functional networks and it has a superior tropism for astrocytes than commonly used serotypes.

The manuscript is well written, the figures are clear and the methods are detailed.

However, there are a number of issues regarding the data that need to be addressed.

- (1) In Suppl fig 2 the authors show a comparison between AAV11 and AAV2-retro on retrograde transduction there appears to be big differences in the populations

transduced with a significant number of cells being transduced only with AAV2 or only AAV11. Would you not expect greater colocalization of the vectors? Is this an artifact of the technique where expression from one vector effects expression from the other in a cell or are they potentially targeting 2 different populations? This is an important consideration if this approach is to be used to efficiently retrogradely transduce a brain region.

Response: Thanks very much for the reviewer's comments and kind suggestion. We injected AAV11-EGFP and AAV2-retro-mCherry in equal amounts into the CPu and found that AAV11 and AAV2-retro could retrogradely co-label the upstream brain regions of the CPu (Supplementary Fig. 2). Even though the in situ diffusion range of AAV11 was reduced by reducing the injection dosage, AAV11 still showed strong retrograde targeting of projection neurons in MD (Supplementary Fig. 2c and d). That is, AAV11 and AAV2-retro exhibit differential infection tropism in the central nervous system of mice. Since the tropism of viruses is mainly determined by their receptors, we believe that these two viruses recognize different receptors and can infect different cell populations, resulting in less colocalization of two vectors in some brain regions.

- (2) For the data on astrocyte transduction there does appear to be a greater number of GFAP positive cells transduced for AAV11 there are cells that are transduced that are not GFAP positive. If the aim is to target astrocytes for gene delivery in disease, specificity is also likely to be crucial. The authors need to present data on what percentage of transduced cells are GFAP positive, what percentage of GFAP positive cells are transduced, and what the non-GFAP labeled cells are transduced.

Response: Thanks very much for the reviewer's comments and kind suggestion. To evaluate the cell-type specificity of the AAV11-GfaABC1D, we measured the colocalization of EGFP and astrocyte marker GFAP (Fig. 6b and c). The results demonstrate that the AAV11-GfaABC1D driven EGFP expression was mainly distributed in the astroglia (Average: 88%, Fig. 6b and c).

- (3) While this paper shows some compelling data regarding the ability to retrogradely label brain areas it would have been nice to see some data showing how this could be used to monitor changes due to disease. For example, modifying the functional networks to show a change in calcium signaling or changes in connectivity due to damage to a specific brain area.

Response: Thanks very much for the reviewer's comments and kind suggestion. Analyzing the differences of the brain neural networks connection between the healthy and diseased animals is the basis of understanding the mechanism of brain diseases caused by neural network abnormalities. To evaluate whether AAV11 can be used for dissecting the variation of circuit connections in Alzheimer's disease mouse brains, AAV11 was injected into the dorsal hippocampus (dHPC) area of APP/PS1 (AD) transgenic mice or wild-type (WT) mice (C57BL/6) (Fig. 7a). Three weeks after

injection, AAV11 could label neurons at the injection site and the upstream brain regions (Fig. 7b and c). Through quantitative analysis, we found that significantly fewer projection neurons in upstream areas, including the contralateral dHPC (Cont-dHPC), entorhinal cortex (ENT) and medial septal complex (MSC), were labelled by AAV11 in APP/PS1 (AD) than those in the wildtype C57BL/6 mice (Fig. 7d). These results indicate that the circuit connections in ten-month-old Alzheimer's disease mouse was decreased significantly compared to the control mice. Thus, AAV11 can be used as a valuable tool for analyzing neural network variation.

Reviewer #2 (Remarks to the Author):

Summary

Retrograde transduction is a feature of many viral vectors, particularly AAVs. Neuroscientists leverage retrograde transduction to study circuit connectivity and to deliver sensors, reporters, and effectors (opsins and DREADDs) to populations of cells that send projections to specific brain regions. AAVs are particularly useful for these studies because they can mediate long-term, strong expression with less neurotoxicity than many other viral vectors. Previously, researchers developed AAV2-Retro, a modified AAV capsid that has improved retrograde transport properties when injected into the rodent or primate brain. This capsid has become commonly used (cited over 700 times) due to the need for efficient retrograde vectors. Here, Han et al. demonstrate the novel finding that a less well studied natural AAV serotype, AAV11 has significant retrograde transduction properties in the mouse brain. The authors compare AAV11 and AAV2-Retro head-to-head in several interconnected brain regions and demonstrate that AAV11 outperforms AAV2-Retro in the transduction of several projection neuron populations (e.g., the CEA and L6 CTX cells after injection into the PAG). They show retrograde transduction with AAV11 paired with Vgat-Cre transgenic mice to label GABAergic projection neurons that send projections to the NAc and they demonstrate, using fiber photometry experiments, that expression from AAV11 is sufficient to detect neuronal activity in VTA neurons projecting to the NAc. Finally, they also demonstrate that AAV11, like many AAVs, transduces astrocytes, a finding that is incomplete and off-topic for this otherwise focused study.

Major concerns

- (1) The authors have not commented on the possible increased local spread of AAV11 vs AAV2-Retro at the injection site (which can be seen Fig 4 and Supplementary Fig 2). Can the increased transduction observed in select neuronal populations be related to the fact that those populations innervate regions surrounding the injection site? The extent of local spread is an important consideration for retrograde labeling experiments, and there are use cases for vectors with wide or limited spread.

Additional characterization of the extent of local spread is an important part of the characterization of AAV11 relative to AAV2-Retro that is lacking in this study.

Response: Thanks very much for the reviewer's comments and kind suggestion. We injected AAV11-EGFP and AAV2-retro-mCherry in equal amounts into the CPu and found that AAV11 and AAV2-retro could retrogradely co-label the upstream brain regions of the CPu (Supplementary Fig. 2). Even though the in situ diffusion range of AAV11 was reduced by reducing the injection dosage, AAV11 still showed strong retrograde targeting of projection neurons in MD (Supplementary Fig. 2c and d). When injected into the vHPC area, AAV11 has higher retrograde transport capacity than AAV2-retro the endopiriform nucleus, dorsal part (EPd, $p = 0.0015$) (Fig. 2e), even though the injection dosage of AAV11 at vHPC was greatly reduced, it could still retrograde target the projection neurons in EPd, while AAV2-retro mainly targeted the projection neurons in piriform area (PIR) (Supplementary Fig. 8). Therefore, the main reason for the difference in infection tropism between the two viruses is not in situ diffusion.

- (2) The abstract highlights AAV11 as a capsid that “has greatly enhanced efficiency over AAV2-Retro”. While true in several specific cases, this appears to be an overstatement. The authors found that AAV2-Retro also outperformed AAV11 in several populations. This manuscript could have the most impact on the field if it gave researchers a thorough and unbiased quantitative comparison between AAV11 and AAV2-Retro.

Response: Thanks very much for the reviewer's comments and kind suggestion. The sentence "AAV11 has greatly enhanced efficiency over AAV2-Retro" was corrected as "AAV11 can function as a powerful retrograde viral tracer complementary to AAV2-retro".

Specific concerns

- (1) The authors repeatedly reference the potential therapeutic use cases for circuit manipulation gene therapy without citing specific examples. It seems premature to be promoting this as a current use case for AAV11 or other vectors.

Response: Thanks very much for the reviewer's comments and kind suggestion. We rechecked and modified the references.

- (2) The authors cite ref 26 in a statement about rational design in the introduction about retrograde vectors. This is an inappropriate citation: Ref 26 is a study on IV administered vectors unrelated to retrograde transport. This work is also cited later for rational design, but this study used panning of a phage display library to identify peptides that were then transferred to the AAV capsid, therefore it is not an example of rational design.

Response: Thanks very much for the reviewer's comments and kind suggestion. The citation was removed.

- (3) The authors refer to different types of AAVs as pedigrees. Clade or serotype would be a more appropriate description depending on whether the authors seek to distinguish between AAVs based on sequence similarity or differential resistance to antibodies.

Response: Thanks very much for the reviewer's comments and kind suggestion. The word "pedigrees" was corrected as "clades".

- (4) At the end of the introduction, the authors speculate about the potential use of AAV11 as a base capsid that could be further engineered to improve its properties. This text should be removed from the introduction. This is an inappropriate place for this, as it will confuse the reader about what was actually done in the study.

Response: Thanks very much for the reviewer's comments and kind suggestion. The sentence at the end of the introduction was removed.

- (5) The entire first paragraph of the results section describes the author's choice not to use the baculovirus system for AAV11 production and to move the AAV11 cap gene into a plasmid for triple transfection. As the later is the standard approach for making research scale AAV, this section seems unnecessary and is better described in the methods, if at all. The only important statement is in the final sentence of the paragraph where the authors describe production yield data.

Response: Thanks very much for the reviewer's comments and kind suggestion. The entire first paragraph of the results section was deleted and only the sentence "For the production of AAV11 vectors, we established a triple plasmid system for AAV11 packaging. Subsequently, the efficiency of virus packaging was evaluated in HEK-293T cells. We found that the pAAV2/11 plasmid could be used to package high-titer AAV11 and the yields of viral particles was equivalent to those of AAV9 (Supplementary Fig. 1)." was retained.

- (6) One well studied population that AAV2-Retro has been reported to target poorly are dopaminergic neurons in the SNc. Given that the authors injected AAV2-Retro and AAV11 into the CPu, they should include a comparison of the transduction of this population or alternatively assess whether their NAc injections which provided expression in the VTA work better with AAV11 than AAV2-Retro.

Response: Thanks very much for the reviewer's comments and kind suggestion. We evaluated the transduction efficiency of the AAV11 virus in the substantia nigra pars compacta (SNc) to dorsal lateral striatum (DLS) pathway, and found that its retrograde transduction efficiency was comparable to that of AAV2-retro (Supplementary Fig. 3b). Both AAV11 and AAV2-retro show low transduction efficiency in the SNc to DLS pathway.

- (7) Previously the authors reported AAV9-Retro, a capsid they generated by transferring the AAV2-Retro insertion to the same region of the AAV9 capsid. Have the authors compared AAV11 with AAV9-Retro?

Response: Thanks very much for the reviewer's comments. Our previous research indicate that rAAV9-Retro and rAAV2-Retro have similar retrograde infection tropism and efficiency at different brain regions. Therefore, we did not compare AAV11 with AAV9-Retro.

- (8) In the section on anterograde transport, the final conclusion is overstated, “These results indicate that AAV11 doesn’t spread anterograde”. The authors should qualify this statement by saying it does not provide anterograde transduction in the tested circuit.

Response: Thanks very much for the reviewer's comments and kind suggestion. The sentence "These results indicate that AAV11 doesn’t spread anterograde" was corrected as "...indicating that AAV11 does not spread anterograde across synapses in primary visual cortex".

- (9) In the results, the authors cite a paper (reference 38) for the use of AAV2-Retro for therapeutic evaluations. However, this paper used AAV2-Retro to express a partial Huntingtin gene fragment (modeling HD, rather than testing a potential therapeutic).

Response: Thanks very much for the reviewer's comments and kind suggestion. The word "therapeutic evaluation" was removed.

- (10) The astrocyte studies are weak and distract from the primary finding of the paper. If the authors wish to include these findings, it would be important to investigate this finding in more depth. Is this finding true across multiple brain regions? The authors cite ref 30, stating that there is variability in the transduction of astrocytes across brain regions by different AAV vectors. Notably, in this prior study, AAV5, which the authors compare with AAV11, underperforms AAV8 in all brain regions, and particularly in the hippocampus (the location the authors chose to examine in the current study). This calls into question the importance and generalizability of their finding. How does AAV11 compare with the more potent AAV8 vector? Does AAV11 provide more efficient transduction of astrocytes in the striatum, a region where AAV5 was more effective? Furthermore, in Fig 7 they inappropriately use $n=27$ (number of sections examined) rather than an $n = 3$ (number of mice examined). As injection site difference can be a major source of variability, this inflates the n and possibly the apparent statistical significance of the findings. As astrocyte transduction is not the primary focus of the paper, the authors could alternatively consider describing the astrocyte transduction (so future users are aware) without making quantitative claims.

Response: Thanks very much for the reviewer's comments and kind suggestion. To evaluate the transduction efficiency of AAV11 on astrocytes, we compared the number of labelled astrocytes after infusion of AAV11 and AAV8 into the right and left sides of the dorsal hippocampus (dHPC), respectively (Fig. 6a and b). Both viral vectors carry an EGFP reporter under the control of the astrocyte-specific truncated GfaABC1D

promoter. To evaluate the cell-type specificity of the AAV11-GfaABC1D, we measured the colocalization of EGFP and astrocyte marker GFAP (Fig. 6b). The results demonstrate that the AAV11-GfaABC1D driven EGFP expression was mainly distributed in the astroglia (Fig. 6b and c). Remarkably, 21 days post-injection, the EGFP- and GFAP-double positive cell counts were significantly higher on the AAV11 side than on the AAV8 side (Fig. 6d), implying that AAV11 has a higher potency in targeting astrocytes than AAV8. We also compared the transduction efficiency of AAV11 and AAV5 on astrocytes in the CPu and dHPC (Supplementary Fig. 10a-d). Quantitative analysis showed that AAV11 was more efficient than AAV5 in targeting astrocytes (Supplementary Fig. 10e and f). Taken together, our study demonstrated that AAV11 is a novel serotype with high astrocyte tropism in vivo. In quantitative analysis, "n=27 (number of sections examined)" was corrected as "n = 3 (number of mice examined)".

- (11)Supplementary Fig 4 has the title “AAV11 for specifically targeting astrocytes”. However, the authors do not show any data or quantitative results demonstrating that AAV11 is specifically targeting astrocytes, nor is this consistent with their other retrograde data. AAV11 is transducing neurons and glia, but they are using a GfABC1D promoter to limit expression to astrocytes. The authors also do not present any quantitative data showing what fraction of the GFP+ cells are astrocytes.

Response: Thanks very much for the reviewer's comments and kind suggestion. The title "AAV11 for specifically targeting astrocytes" was corrected as "AAV11 combined with AAV1 for analyzing neuron-astrocyte circuit connection". To evaluate the cell-type specificity of the AAV11-GfaABC1D, we measured the colocalization of EGFP and astrocyte marker GFAP (Fig. 6b). The results demonstrate that the AAV11-GfaABC1D driven EGFP expression was mainly distributed in the astroglia (Fig. 6b and c). AAV1 based anterograde axoastrocytic transfer can be used to study neuron-astrocyte circuit connection (Reference 41). Given AAV11 can transduce projection neurons and astrocytes with high efficiency, we hypothesized that AAV11 can be used together with AAV1 to analyze neuron-astrocyte circuit connection. To test this hypothesis, we coinjected AAV1-CMV-Cre and AAV5-EF1 α -fDIO-tdTomato into the ventral posterior medial nucleus (VPM), and AAV11-GfaABC1D-DIO-EGFP and AAV11-EF1 α -DIO-Flp into the barrel cortex (BX) (Fig. 6e). Representative images demonstrated that only VPM neurons projected to BX were marked by tdTomato (red), and BX astrocytes connected with VPM neurons were marked by EGFP (green) (Fig. 6f). Therefore, AAV11 can be used in combination with AAV1 to analyze neuron-astrocyte circuit connection.

- (12)Given that the true variable the authors are seeking to study is AAV11 vs AAV2-Retro, and given that virus preps can vary significantly from lot to lot, it is imperative that the authors comment on how many independently produced AAV11 and AAV2-Retro lots were compared in this study. If all of these comparisons were from a single lot of either vector, the authors should evaluate and report results

from additional lots of both vectors in at least a subset of these assays to ensure reproducibility of their findings.

Response: Thanks very much for the reviewer's comments and kind suggestion. When injected into the vHPC area (Fig. 2a), AAV2-retro was more prone to retrograde infect the lateral entorhinal cortex (ENT), with a significantly higher transduction efficiency than that of AAV11 ($p = 0.0205$, Fig. 2e). In contrast, AAV11 has higher retrograde transport capacity than AAV2-retro in other upstream brain regions including the hippocampal region (dHPC, $p = 0.0033$), the midline group of the dorsal thalamus (MTN, $p = 0.0048$), the endopiriform nucleus, dorsal part (EPd, $p = 0.0015$), the lateral dorsal nucleus of thalamus (LD, $p = 0.0027$), and the medial septal complex (MSC, $p = 0.0357$) (Fig. 2e). Importantly, similar effects were observed in the experimental groups of separate injection (Supplementary Fig. 5), different virus batches (Supplementary Fig. 6) and color exchange (Supplementary Fig. 7). Even though the injection dosage of AAV11 at vHPC was greatly reduced, it could still retrograde target the projection neurons in EPd, while AAV2-retro mainly targeted the projection neurons in piriform area (PIR) (Supplementary Fig. 8).

(13) The authors claim in their abstract and discussion that they showed the potential of AAV11 to monitor activities in a functional network using a cell-type-specific promoter, but used hSyn as the promoter for this study. This promoter is broadly expressed in nearly all neuron populations, so this claim is overstated.

Response: Thanks very much for the reviewer's comments and kind suggestion. The word "cell-type-specific" was corrected as "neuron-specific".

(14) The authors have not provided any description of their cell counting methods or the statistical tests used. The statistical tests used should also be described in each relevant figure legend.

Response: Thanks very much for the reviewer's kind suggestion. For cell counting, one-sixth of brain slices from each animal were selected. Positive cells were quantified using ImageJ software v1.8.0 (National Institutes of Health, Bethesda, MD, USA). Data were shown as means \pm standard error of the mean (SEM) using GraphPad Prism7.

Reviewer #3 (Remarks to the Author):

This manuscript by Han et al reports the use of naturally occurring AAV11 serotype for retrograde labeling of neurons, which allows for neuronal circuit interrogation in the CNS. Overall, this is an interesting finding, however, the finding is only incremental in nature when one considers that AAV2-retro has already been developed for retrograde labeling of circuits. Many of the findings reported in this manuscript suggest that AAV11 is an optimized version of AAV2-retro, with the ability to detect a larger number

of retrogradely labeled neurons or the ability to label a few difficult to label circuits in the brain. In addition to the points above that relate to the impact and significance of this study, there are major points and other points that will need to be considered by the authors for the manuscript. These are listed below:

MAJOR POINTS:

- (1) The data on astrocytes (Fig. 7) appear to be completely out of context with the manuscript that has the goal of examining AAV11 as a retrograde reporter virus. In addition, data shown in Fig 7d suggest a very modest increase in the ability of AAV11 to label astrocytes compared to the well used and well characterized AAV5 serotype. The impact of this manuscript will be greatly increased if the authors can show that AAV11 can label axonal processes within the territory of transduced astrocytes, which would be a very useful and impactful tool for the field. In fact a recent study (PMID: 35138891) has shown axoastrocytic AAV transfer using novel methods. The authors could use similar methods to examine if AAV11 is capable of astroaxonal transfer.

Response: Thanks very much for the reviewer's comments and kind suggestion. AAV1 based anterograde axoastrocytic transfer can be used to study neuron-astrocyte circuit connection (Reference 41). Given AAV11 can transduce projection neurons and astrocytes with high efficiency, we hypothesized that AAV11 can be used together with AAV1 to analyze neuron-astrocyte circuit connection. To test this hypothesis, we coinjected AAV1-CMV-Cre and AAV5-EF1 α -fDIO-tdTomato into the ventral posterior medial nucleus (VPM), and AAV11-GfaABC1D-DIO-EGFP and AAV11-EF1 α -DIO-Flp into the barrel cortex (BX) (Fig. 6e). Representative images demonstrated that only VPM neurons projected to BX were marked by tdTomato (magenta), and BX astrocytes connected with VPM neurons were marked by EGFP (green) (Fig. 6f). Therefore, AAV11 can be used in combination with AAV1 to analyze neuron-astrocyte circuit connection.

- (2) All figures show very low resolution / magnification images of cell bodies. It is important to show zoomed in confocal images of the cell bodies and quantify cell bodies labeled for AAV2-retro and AAV11 in this way. This is especially important because the quantal yield of mcherry is several fold lower than GFP, which can easily lead to a wrong interpretation that the labeling efficiency of AAV2-retro is lower than that of AAV11. This fact holds true for all the data shown in this manuscript.

Response: Thanks very much for the reviewer's comments and kind suggestion. According to reviewer's suggestion, imaging was performed using the Leica TCS SP8 confocal microscope (Leica, Wetzlar, Germany).

- (3) It is unclear how the use of AAV11 as shown in Figure 6 provides advantages over existing methods to label neurons with GCaMPs. For instance it is easily possible to directly label the neuronal nucleus of interest using the multiple Cre mouse lines.

The authors need to design and perform these experiments in a way that truly bring out the advantage of using AAV11 over AAV2-retro in the context of interrogating functional circuits.

Response: Thanks very much for the reviewer's comments and kind suggestion. Our work indicate that AAV11 can function as a powerful retrograde viral tracer complementary to AAV2-retro. Then, as a method verification, we choose a known neural pathway responsible for reward to verify whether AAV11 can be used for functional network analysis. Therefore, we did not compare AAV11 and AAV2-retro in this section.

OTHER POINTS:

(1) Title: The use of the word "performs" in the title is odd. Please revise the title.

Response: Thanks very much for the reviewer's comments and kind suggestion. The word "performs" was corrected as "enables".

(2) Line 48: The word "mental" should be "neurological"

Response: Thanks very much for the reviewer's kind suggestion. The word "mental" was corrected as "neurological".

(3) Line 60: Please add references for each of the viral vector types mentioned in this sentence

Response: Thanks very much for the reviewer's kind suggestion. According to the reviewer's kind suggestion, we added references for each of the viral vector types

(4) Supplementary Fig 1: Titers are quite low. The methods do not details if and how a large scale AAV prep was made. Did the authors use a 10-layer cell culture system, and how was the vector prep purified. These details need to be mentioned in the methods.

Response: Thanks very much for the reviewer's comments and kind suggestion. Supplementary Fig. 1 shows the average yield of AAV per HEK-293T cell, rather than the titer of concentrated AAV. Each AAV was prepared using 15 cell culture dishes (100 × 20 mm). Viral particles were purified by iodixanol gradient ultracentrifugation.

(5) Line 107: The word "infectious effect" is ambiguous and odd. Please rephrase this line.

Response: Thanks very much for the reviewer's kind suggestion. The word "infectious effect" was corrected as "transduction effect".

(6) Fig 1a and Fig 2 need to show labeling of the SNc and VTA, which are major nuclei projecting to the NAc for both retro-AAV2 and AAV11. Confocal microscopy with high resolution imaging of the SNc and VTA would be necessary here.

Response: Thanks very much for the reviewer's comments and kind suggestion. We evaluated the transduction efficiency of the AAV11 virus in the substantia nigra pars compacta (SNc) to dorsal lateral striatum (DLS) pathway, and found that its retrograde transduction efficiency was comparable to that of AAV2-retro (Supplementary Fig. 3b). Both AAV11 and AAV2-retro show low transduction efficiency in the SNc to DLS pathway.

- (7) Supplementary Fig 3: One important control here is to use AAV2-mcherry (non-retro version) as a negative control. Sections that are 2.2, 0.5, -0.5 mm from bregma look different in many regions. This experiment needs to be repeated with dual color labeling in the same animal (AAV2-retro-mcherry + AAV11 GFP).

Response: Thanks very much for the reviewer's comments and kind suggestion. We injected AAV11-EGFP and AAV2-retro-mCherry in equal amounts into the CPu and found that AAV11 and AAV2-retro could retrogradely co-label the upstream brain regions of the CPu (Supplementary Fig. 2). Even though the in situ diffusion range of AAV11 was reduced by reducing the injection dosage, AAV11 still showed strong retrograde targeting of projection neurons in MD (Supplementary Fig. 2c and d). That is, AAV11 and AAV2-retro exhibit differential infection tropism in the central nervous system of mice.

- (8) Line 126: "doesn't" needs to be "does not"

Response: Thanks very much for the reviewer's kind suggestion. The word "doesn't" was corrected as "does not".

- (9) Fig 5c needs to show a head-to-head comparison with AAV2-retro-mcherry virus using the DIO flex system in AAV2-retro

Response: Thanks very much for the reviewer's kind suggestion. We further compared the efficiency between AAV11 and AAV2-retro in retrograde transduction of GABAergic neurons projecting to NAc (Fig. 4a). We found that there are different tracing effects in the upstream brain regions between two viruses (Fig. 4b). Quantitative analysis demonstrated that AAV11 was more efficient than AAV2-retro at retrograde labelling of the striatum-like amygdalar nuclei (sAMY, $p = 0.0030$) and the periaqueductal gray (PAG, $p = 0.0295$) (Fig. 4c), while AAV2-retro was more efficient than AAV11 at retrograde labelling of the ventral tegmental area (VTA, $p = 0.0256$). In addition, there were similar transduction efficiencies between two serotypes in the dorsomedial nucleus of the hypothalamus (DMH, $p = 0.8956$) (Fig. 4c). These results demonstrate that dissection of genetically defined projection neurons can be implemented by combining AAV11 with available transgenic lines (Cre or Flp).

- (10) Line 173: "GCaMP detects calcium flux at the scale of single action potentials". This sentence is wrong. GCaMPs do not have the resolution to detect single action potential calcium fluxes. The data reported by the authors are very likely to be because of burst firing of neurons. Please correct this sentence.

Response: Thanks very much for the reviewer's comments and kind suggestion. The sentence "GCaMP detects calcium flux at the scale of single action potentials" was corrected as "GCaMP can reliably monitor neuronal calcium transients".

(11) Discussion: There needs to be more discussion on the mechanism, receptors etc that may be involved in retrograde infection of axonal terminals.

Response: Thanks very much for the reviewer's comments and kind suggestion. Most AAVs require both GPR108 and AAVR (KIAA0319L) receptors for cellular entry, except for AAV4 and rh32.33, which are GPR108-dependent yet AAVR-independent (Reference 50-52). Since both AAV11 and rh32.33 belong to AAV4-clade members (Reference 18), AAV11 may be dependent on GPR108 for successful cellular transduction. The next important work is to pinpoint the possible binding receptors through single cell sequencing.

(12) Methods: The methods section lacks a description of how the neuronal cell bodies were quantified in this study. Did the authors use ImageJ or some other program? Were the cell bodies thresholded for analysis?

Response: Thanks very much for the reviewer's kind suggestion. For cell counting, one-sixth of brain slices from each animal were selected. Positive cells were quantified using ImageJ software v1.8.0 (National Institutes of Health, Bethesda, MD, USA). Data were shown as means \pm standard error of the mean (SEM) using GraphPad Prism 7.0 or Origin 7.0 (OriginLab, Northampton, MA, USA).

REVIEWER COMMENTS

Reviewer #1 (Remarks to the Author):

Thank you to the authors for the revised manuscript and for addressing my comments.

Regarding my question about the lack of colocalisation in suppl. Fig 2, the authors have commented that the two viruses recognize different receptors and can infect different cell populations, resulting in less colocalization of two vectors in some brain regions, which is, of course, correct. However, I feel it is important to understand what are the different populations being transduced. Are they different populations of the same cell type or two distinct cell types? This is crucial to understand if this vector is to be used for mapping and manipulating neural circuits and for gene therapy of some neurological and neurodegenerative disorders as the authors suggest.

Reviewer #2 (Remarks to the Author):

The authors have largely addressed my concerns with new experiments and edits to the manuscript.

However a few mostly minor details should be corrected:

1. In several places include the abstract and line 91, the authors use the phrase axoastrocytic transfer technology, which is rather ambiguous. It may be preferable to describe this as axoastrocytic labeling methods or bidirectional, multi-vector axoastrocytic labeling.
2. Their experiment looking at retrograde labeling efficiencies in the AD mouse model is a helpful new addition to the paper, but the authors make a very definitive statement that these results indicate that the circuit connections in the APP/PS1 animals were decreased significantly. While this may be true, their experiment does not provide proof of this as there could be many other causes of the reduced labeling in these mice. For example, reduced transgene expression, altered immune response to the AAV, or more inefficient retrograde transport.
3. The authors have not listed the statistical tests they used to generate p values in the figure legend.
4. The authors did not directly address the question I originally raised in my prior review (Major Concern 1) that the differential transduction observed with AAV2-retro and AAV11 is due to differential spread by AAV11 and AAV2. Spread is a property of capsid-receptor interactions and biophysical properties rather than dosage, so while their new dilution series data is helpful, it does not address whether the differential retrograde transduction by the AAVs is related to the vectors spreading into different axon terminal fields even when co-injected. I do not think additional experiments are necessary, but it would be helpful given that so little is known about the use of AAV11 in the brain, to show examples of individual channel, low magnification images of the injection sites as a supplementary figure so readers

can better judge the diffusion characteristics of the two vectors.

5. Independently, their dilution series experiment is helpful addition, but the authors do not describe the rationale for this experiment nor the results in any detail in the manuscript. The observation that appears striking to this reviewer is that there is minimal overlap between cortical cells transduced by AAV11 and AAV2-retro after injection into the striatum. Each vector is biased toward different cortical layers, which is very surprising. The authors should discuss this interesting observation. Again, AAV11 may have important use cases, not just as a general better performing retrograde vector than AAV2-retro, but a possible complementary vector for labeling distinct subsets of cells.

6. The authors describe searching for AAV capsids with better retrograde properties by looking for new serotypes as a more fundamental approach (lines 76-77). It would be more appropriate to state that this is an alternative approach to capsid engineering.

7. (Lines 78-79) the authors discuss possible future engineering of AAV11 or other natural serotypes in the introduction. This speculative statement without citation is not appropriate for the introduction.

8. Because they are not working with replication competent viruses, the authors should use the word transduction instead of infection (e.g, line 138).

9. The new addition to Fig 6 (e and f), is a nice addition to the paper that helps to tie together the retrograde and astrocyte tropism data. However, given the relatively high rate of Cre/Flp independent recombination that can occur in recombinase-dependent vectors during plasmid amplification and AAV production, it is important to show that the recombinase-dependent vectors have low or no recombination in the absence of the injection of the Cre and Flp vectors using the same virus stocks, doses, and injection parameters.

10. The authors should add titers and volume injections to Supplementary Fig. 6.

Reviewer #3 (Remarks to the Author):

The authors have performed a comprehensive revision of the manuscript based on comments. As shown in Figure 6 of the revised manuscript, AAV11 does not seem to retrogradely express in axons within the cortex. Additionally, the dual injection strategy used here does not really provide any advantages over existing methods. The significance of this manuscript is limited to the natural ability of AAV11 for retrograde transduction, and the data on increased astrocyte tropism compared to other AAV serotypes are not convincing. In fact, the authors use AAV8 for comparison instead of AAV5, which is the AAV serotype most often used for astrocyte expression. Therefore, the comparison shown in Figure 6d needs to be redone with AAV5 as the reference rather than AAV8.

Responses to Reviewers' Comments

We are grateful for the reviewer's comments and suggestions. In response to the reviewers' comments, we respectfully provide our point-by-point response below (marked as blue).

Reviewer #1 (Remarks to the Author):

Thank you to the authors for the revised manuscript and for addressing my comments.

Regarding my question about the lack of colocalization in suppl. Fig 2, the authors have commented that the two viruses recognize different receptors and can infect different cell populations, resulting in less colocalization of two vectors in some brain regions, which is, of course, correct. However, I feel it is important to understand what are the different populations being transduced. Are they different populations of the same cell type or two distinct cell types? This is crucial to understand if this vector is to be used for mapping and manipulating neural circuits and for gene therapy of some neurological and neurodegenerative disorders as the authors suggest.

Response: We sincerely appreciate the reviewer's comments and valuable suggestions. In response, we identified the differences of neuronal cell types labelled by two viral vectors in the SSp brain region using fluorescent in situ hybridization and immunofluorescence methods (Supplementary Fig. 13).

Both AAV11 and AAV2-retro labelled mostly excitatory neurons (~90%), but AAV11 marked more inhibitory neurons than AAV2-retro in the SSp area (Supplementary Fig. 13c). This efficient transduction of inhibitory neurons by AAV11 was also confirmed in *Vgat-ires-Cre* mice (Fig. 4).

However, it is important to note that the tropism of a specific vector for a certain type of neuron in a particular brain area should not be assumed to be applicable to other brain areas. At present, we are inclined to believe that AAV11 shows differential transduction tropism towards cell types in different circuits (e.g., Fig. 4), and the underlying mechanisms for this phenomenon require further investigation into the transduction mechanism of AAV11.

Reviewer #2 (Remarks to the Author):

The authors have largely addressed my concerns with new experiments and edits to the manuscript.

However, a few mostly minor details should be corrected:

1. In several places include the abstract and line 91, the authors use the phrase axoastrocytic transfer technology, which is rather ambiguous. It may be preferable to describe this as axoastrocytic labeling methods or bidirectional, multi-vector axoastrocytic labeling.

Response: Thanks very much for the reviewer's comments and kind suggestion. We have revised "(anterograde) axoastrocytic transfer technology" to "bidirectional multi-vector axoastrocytic labeling" in the manuscript as you recommended.

2. Their experiment looking at retrograde labeling efficiencies in the AD mouse model is a helpful new addition to the paper, but the authors make a very definitive statement that these results indicate that the circuit connections in the APP/PS1 animals were decreased significantly. While this may be true, their experiment does not provide proof of this as there could be many other causes of the reduced labeling in these mice. For example, reduced transgene expression, altered immune response to the AAV, or more inefficient retrograde transport.

Response: Thanks very much for the reviewer's comments and kind suggestion. We have revised this section in the manuscript accordingly.

3. The authors have not listed the statistical tests they used to generate p values in the figure legend.

Response: Thanks very much for the reviewer's comments and kind suggestion. All statistical analyses were performed using the software Graphpad Prism and p values were computed by unpaired two sample t tests. We have revised the manuscript accordingly.

4. The authors did not directly address the question I originally raised in my prior review (Major Concern 1) that the differential transduction observed with AAV2-retro and AAV11 is due to differential spread by AAV11 and AAV2. Spread is a property of capsid-receptor interactions and biophysical properties rather than dosage, so while their new dilution series data is helpful, it does not address whether the differential retrograde transduction by the AAVs is related to the vectors spreading into different axon terminal fields even when co-injected. I do not think additional experiments are necessary, but it would be helpful given that so little is known about the use of AAV11 in the brain, to show examples of individual channel,

low magnification images of the injection sites as a supplementary figure so readers can better judge the diffusion characteristics of the two vectors.

Response: We sincerely appreciate the reviewer's comments and valuable suggestions. In response, we imaged the injection sites (both CPu and vHPC) using confocal microscopy (Supplementary Fig. 14). It is worth noting that the diffusion range of AAV11 is indeed higher than that of AAV2-retro at the CPu injection site, which may be one of the reasons for their differential effects on projecting neurons labeling. Although vHPC could limit the diffusion range of both viral vectors to almost the same extent due to its unique structural characteristics, differences in the transduced cell populations between the two vectors were still observed. This further confirms the differential transduction tropism of the two viral vectors towards neurons.

5. Independently, their dilution series experiment is helpful addition, but the authors do not describe the rationale for this experiment nor the results in any detail in the manuscript. The observation that appears striking to this reviewer is that there is minimal overlap between cortical cells transduced by AAV11 and AAV2-retro after injection into the striatum. Each vector is biased toward different cortical layers, which is very surprising. The authors should discuss this interesting observation. Again, AAV11 may have important use cases, not just as a general better performing retrograde vector than AAV2-retro, but a possible complementary vector for labeling distinct subsets of cells.

Response: We appreciate the reviewer's comments and suggestions. We identified the differences of neuronal cell types labelled by two viral vectors in the SSp brain region using fluorescent *in situ* hybridization and immunofluorescence methods and further discussed this phenomenon (Supplementary Fig. 13, line 333-348).

As you mentioned, AAV11 is not just a generally better performing retrograde vector than AAV2-retro, but also a possible complementary vector for labeling distinct subsets of cells. We further confirmed this by detecting the *in situ* transduction (Supplementary Fig. 14).

Apart from both AAV11 and AAV2-retro viruses being able to label some upstream cell populations, the differences in the effect of their retrograde labeling can be divided into the following three situations:: 1) AAV11 can target certain brain areas, while AAV2-retro is rare (e.g., tagging MSC from vHPC, Fig. 2d); 2) Conversely, AAV2-retro can target certain brain areas, while AAV11 is rare (e.g., tagging ENT or PIR from vHPC, Fig. 2e, Supplementary Fig. 5); 3) AAV11 and AAV2-retro can both label a certain upstream brain area, but they exhibit different neuronal subpopulation targeting (e.g., tagging SSp from CPu, Supplementary Fig. 5).

At present, we tend to believe that AAV11 shows differential transduction tropism towards cell types in different circuits (e.g., Fig. 4), and further investigation into the

transduction mechanism of AAV11 is required to understand the underlying mechanisms for this phenomenon.

6. The authors describe searching for AAV capsids with better retrograde properties by looking for new serotypes as a more fundamental approach (lines 76-77). It would be more appropriate to state that this is an alternative approach to capsid engineering.

Response: Thanks very much for the reviewer's comments and kind suggestion. We have revised the wording in the manuscript.

7. (Lines 78-79) the authors discuss possible future engineering of AAV11 or other natural serotypes in the introduction. This speculative statement without citation is not appropriate for the introduction.

Response: Thanks very much for the reviewer's comments and kind suggestion. We have removed the corresponding content from the manuscript.

8. Because they are not working with replication competent viruses, the authors should use the word transduction instead of infection (e.g, line 138).

Response: Thanks very much for the reviewer's comments and kind suggestion. We have revised the manuscript and replaced "infection" with "transduction" accordingly.

9. The new addition to Fig 6 (e and f), is a nice addition to the paper that helps to tie together the retrograde and astrocyte tropism data. However, given the relatively high rate of Cre/Flp independent recombination that can occur in recombinase-dependent vectors during plasmid amplification and AAV production, it is important to show that the recombinase-dependent vectors have low or no recombination in the absence of the injection of the Cre and Flp vectors using the same virus stocks, doses, and injection parameters.

Response: Thanks very much for the reviewer's comments and kind suggestion. We have added relevant data to demonstrate that the recombinase-dependent vectors exhibit low or no recombination in the absence of the injection of the Cre and Flp vectors. (Supplementary Fig. 12)

10. The authors should add titers and volume injections to Supplementary Fig. 6.

Response: Thanks very much for the reviewer's comments and kind suggestion. we have added information on viral dosage and injection volume in Supplementary Fig. 6.

Reviewer #3 (Remarks to the Author):

The authors have performed a comprehensive revision of the manuscript based on comments. As shown in Figure 6 of the revised manuscript, AAV11 does not seem to retrogradely express in axons within the cortex. Additionally, the dual injection strategy used here does not really provide any advantages over existing methods. The significance of this manuscript is limited to the natural ability of AAV11 for retrograde transduction, and the data on increased astrocyte tropism compared to other AAV serotypes are not convincing. In fact, the authors use AAV8 for comparison instead of AAV5, which is the AAV serotype most often used for astrocyte expression. Therefore, the comparison shown in Figure 6d needs to be redone with AAV5 as the reference rather than AAV8.

Response: We would like to express our sincere gratitude for the reviewer's comments and valuable suggestions. Given that previous studies have shown that AAV8 has better transduction efficiency of astrocytes than AAV5 in the hippocampus ¹, we compared AAV11 with AAV8. Of course, previous studies have also shown that AAV5 has better transduction efficiency of astrocytes than AAV8 in the CPu. As mentioned by the reviewer, AAV5 is the AAV serotype most often used for astrocyte expression, to further improve this part of our study, we also compared AAV11 with AAV5 in the CPu (Fig 6. c, d and g) and dHPC (Supplementary Fig. 10) regions, respectively.

Reference:

1. Aschauer, D. F., Kreuz S., Rumpel S. Analysis of transduction efficiency, tropism and axonal transport of AAV serotypes 1, 2, 5, 6, 8 and 9 in the mouse brain. *PLoS ONE* **8**, e76310 (2013).

REVIEWERS' COMMENTS

Reviewer #1 (Remarks to the Author):

The authors have addressed my comment - thank you.

Reviewer #2 (Remarks to the Author):

The authors have addressed my concerns. My only remaining comments are to clarify the description of some of the new data.

Starting in line 282, the authors state:

"AAV11 can target certain brain areas, while AAV2-retro is rare" and "Conversely, AAV2-retro can target certain brain areas, while AAV11 is rare"

The authors should clarify the above statements. Taken together the statements provide little information and the two phrases in each statement do not make sense as a comparison. Perhaps they could speak about the relative efficiencies of each vector in the specific locations?

The authors state "AAV11 and AAV2-retro can both label a certain upstream brain area, but they exhibit different neuronal subpopulation targeting"

Again, this sentence conveys little information that is useful to the reader. The use of the word certain and different here is very ambiguous and not helpful. The authors are encouraged to be more descriptive.

Detection of the CPU injection site revealed that AAV11 and AAV2-retro exhibited different diffusion patterns (Supplementary Fig. 14a), which may contribute to their differential retrograde labeling efficacy by spreading into different axon terminal fields

The author's addition of Sup Fig 14 is helpful here. However, their language again could be more descriptive with directional changes. For example, it would be more helpful to the reader to say AAV11 showed greater (not different) diffusion than AAV2-Retro ... spreading into a larger axon terminal field.

Reviewer #3 (Remarks to the Author):

My comment has been addressed by the addition of new data. This paper would be a useful contribution to the field.

Responses to Reviewers' Comments

We are grateful for the reviewer's comments and suggestions. In response to the reviewers' comments, we respectfully provide our point-by-point response below (marked as blue).

Reviewer #1 (Remarks to the Author):

The authors have addressed my comment - thank you.

Response: We appreciate the reviewer's recognition of our revised manuscript.

Reviewer #2 (Remarks to the Author):

The authors have addressed my concerns. My only remaining comments are to clarify the description of some of the new data.

1. Starting in line 282, the authors state:

“AAV11 can target certain brain areas, while AAV2-retro is rare” and “Conversely, AAV2-retro can target certain brain areas, while AAV11 is rare”

The authors should clarify the above statements. Taken together the statements provide little information and the two phrases in each statement do not make sense as a comparison. Perhaps they could speak about the relative efficiencies of each vector in the specific locations?

Response: We appreciate the reviewer's constructive suggestions and have made targeted revisions to the manuscript accordingly. The sentence was corrected as “While AAV11 and AAV2-retro shared labeling traits in some brain regions, their unique characteristics provided complementary transduction advantages elsewhere. For example, in the vHPC, AAV2-retro labeled the ENT or PIR retrogradely more effectively, whereas AAV11 showed a greater efficacy in transducing the MSC or EPd (Fig. 2, Supplementary Fig. 5, Supplementary Fig. 7)”.

2. The authors state “AAV11 and AAV2-retro can both label a certain upstream brain area, but they exhibit different neuronal subpopulation targeting”

Again, this sentence conveys little information that is useful to the reader. The use of the word certain and different here is very ambiguous and not helpful. The authors are encouraged to be more descriptive.

Response: We appreciate the reviewer's constructive suggestions and have made

targeted revisions to the manuscript accordingly. The sentence was corrected as “Notably, in the CPu, both vectors retrogradely labeled the SSp region, but each demonstrated a preference towards different layers of neuronal populations within the region (Supplementary Fig. 2)”.

3. Detection of the CPu injection site revealed that AAV11 and AAV2-retro exhibited different diffusion patterns (Supplementary Fig. 14a), which may contribute to their differential retrograde labeling efficacy by spreading into different axon terminal fields.

The author's addition of Sup Fig 14 is helpful here. However, their language again could be more descriptive with directional changes. For example, it would be more helpful to the reader to say AAV11 showed greater (not different) diffusion than AAV2-Retro ... spreading into a larger axon terminal field.

Response: We appreciate the reviewer's constructive suggestions and have made targeted revisions to the manuscript accordingly. The sentence was corrected as “Detection of the CPu injection site revealed that AAV11 showed greater diffusion than AAV2-retro (Supplementary Fig. 14a), which may contribute to their differential retrograde labeling efficacy by spreading into a larger axon terminal field.”

Reviewer #3 (Remarks to the Author):

My comment has been addressed by the addition of new data. This paper would be a useful contribution to the field.

Response: We appreciate the reviewer's recognition of our revised manuscript.